# PRESCRIBE: Predicting Single-Cell Responses with Bayesian Estimation

**Jiabei Cheng[1], Changxi Chi[2], Jingbo Zhou[2], Hongyi Xin[1]\*, Jun Xia[3,4]\***
[1]Shanghai Jiao Tong University, [2]Westlake University,
[3]AIMS Lab, The Hong Kong University of Science and Technology (Guangzhou),
[4]The Hong Kong University of Science and Technology
`{jiabei_cheng, hongyi.xin}@sjtu.edu.cn`
`junxia@hkust-gz.edu.cn`

## Abstract

In single-cell perturbation prediction, a central task is to forecast the effects of perturbing a gene unseen in the training data. The efficacy of such predictions depends on two factors: (1) the similarity of the target gene to those covered in the training data, which informs model (epistemic) uncertainty, and (2) the quality of the corresponding training data, which reflects data (aleatoric) uncertainty. Both factors are critical for determining the reliability of a prediction, particularly as gene perturbation is an inherently stochastic biochemical process. In this paper, we propose **PRESCRIBE** (**PRE**dicting **S**ingle-**C**ell **R**esponse w**I**th **B**ayesian **E**stimation), a multivariate deep evidential regression framework designed to measure both sources of uncertainty jointly. Our analysis demonstrates that PRESCRIBE effectively estimates a confidence score for each prediction, which strongly correlates with its empirical accuracy. This capability enables the filtering of untrustworthy results, and in our experiments, it achieves steady accuracy improvements of over 3% compared to comparable baselines. Code is available at `https://github.com/Bunnybeibei/PRESCRIBE`.

## 1 Introduction

Predicting the effects of perturbations is crucial for advancing biological understanding and the development of targeted genetic therapies. Recent years have seen significant progress in machine learning models [1–3], data generation [4, 5], and benchmarking [6, 7] for this task. However, a critical challenge remains less explored: quantifying prediction uncertainty for individual predictions, especially for perturbations of genes that are not seen during training and are functionally distant to any of the genes in the training set. Fig. 1(a) illustrates this issue, showing that even models with high average accuracy can make substantial errors on specific predictions.

The first step towards estimating prediction uncertainty is to understand its sources. Prediction uncertainty arises from the interaction between two primary sources. First, **data (aleatoric) uncertainty** arises from the inherent stochasticity of biological systems, where perturbing a single gene can yield a diverse spectrum of cellular outcomes. Second, **model (epistemic) uncertainty** reflects the model's unfamiliarity with a given input, which is particularly high for out-of-distribution perturbations. A practical framework needs to account for both. For example, a prediction is the least reliable when the model's output is far from a highly consistent biological outcome (high model uncertainty, low data uncertainty). Equally, a prediction is intrinsically uncertain when the biological outcome itself is highly variable (high data uncertainty).

---

*\*Corresponding authors.*

39th Conference on Neural Information Processing Systems (NeurIPS 2025).

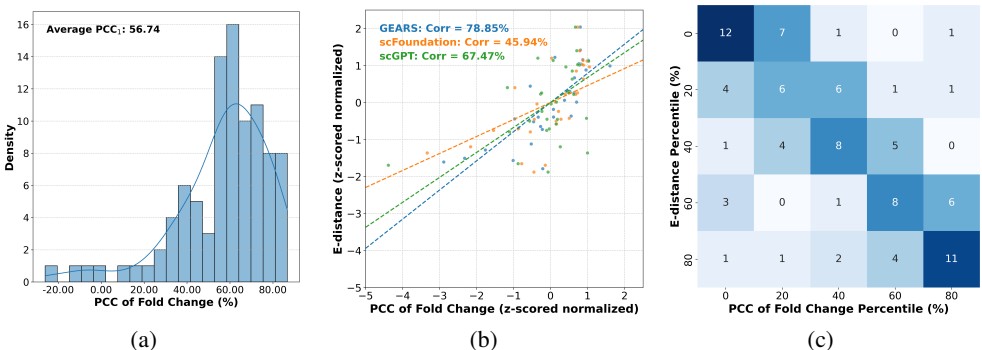

Figure 1: Preliminary experiments on the Norman dataset using three representative models: (a) High overall predictive accuracy does not ensure individual prediction reliability. (b) E-distance exhibits a strong correlation with prediction accuracy across these models. (c) Calibration analysis demonstrates that E-distance percentiles effectively stratify prediction accuracy.

Here, we propose a unified, data-driven metric to capture both data and model uncertainty, inspired by the energy distance (E-distance) [8]. E-distance quantifies the similarity between two cell populations by balancing the distance *between* them against the dispersion *within* them. In the context of perturbation prediction, E-distance reconciles both the model uncertainty and the data uncertainty (Fig. 1(b)). If the ground truth of a perturbation is known a priori, then the E-distance between the predicted post-perturbation population and the ground truth, composed of an inter-group distance compensated by a negative intra-group distance term, reflects the accuracy of the prediction (Fig. 1(c)). Apparently, the ground truth post-perturbation cell fate distribution of an unseen gene cannot be obtained. Therefore, E-distance, in its original form, is not directly applicable to the perturbation prediction uncertainty estimation task. In this work, we adapt the idea of E-distance by jointly modeling the prediction error and the intrinsic post-perturbation dispersion. We name our perturbation uncertainty estimation metric pseudo E-distance.

To estimate the pseudo E-distance for unseen perturbations, we introduce **PRESCRIBE** (**PRE**dicting **S**ingle-**C**ell **R**esponse w**I**th **B**ayesian **E**stimation). PRESCRIBE is a multivariate extension of a deep evidential learning framework (Natural Posterior Network [9]) that, instead of outputting a single expression profile, simultaneously predicts the post-perturbation state and estimates the prediction's confidence. PRESCRIBE has two key elements: a posterior distribution over the transcriptomic landscape and an **evidence score** derived from a learned latent density of the perturbation space. The pseudo E-distance (Fig. 2) is defined by combining these terms: the spread of the predicted distribution (measured by its entropy) quantifies data uncertainty, while the evidence score quantifies model uncertainty. Intuitively, the evidence score measures the distribution density of the training data in the latent perturbation space within a close vicinity of an unseen target gene or gene combination. A high evidence score indicates that a prediction is grounded by multiple functionally related gene perturbation instances in the training data, and a low evidence score indicates otherwise. We model the post-perturbation transcriptomic spread using a Normal-Wishart conjugate Bayesian framework, allowing the variance to be inferred from the perturbation embedding via a trained decoder. For perturbations far from the training data, the predicted distribution defaults to a null state (typically the unperturbed control cell population). This fallback ensures a safe output for unreliable predictions.

Our main contributions can be summarized as follows:

- We propose PRESCRIBE, a novel framework that uses a predicted pseudo E-distance as a unified surrogate for both data and model uncertainty in single-cell perturbation prediction.

- We introduce a multivariate extension of the Natural Posterior Network, utilizing an Inverse-Wishart prior to effectively model predictive distributions over multi-dimensional gene expression states.

- We demonstrate through comprehensive experiments that PRESCRIBE generates well-calibrated uncertainty scores that improve predictive accuracy by enabling the filtering of unreliable results.

## 2 Related Work

### 2.1 Predicting Single-Cell Responses

In-silico prediction of single-cell responses offers an efficient alternative to costly single-cell perturbations in the wet lab. Current methods generally fall into two main categories: direct matching or disentanglement [7]. Direct matching methods, such as GEARS [1] and scGPT [10], map control cell gene expressions to perturbed expressions to predict responses to new perturbations. Disentanglement methods, like CPA [11], isolate perturbation effects from cellular features (e.g., cell type, dosage) to enable predictions under diverse conditions. Our work aligns with the direct matching strategy, focusing on the challenge of predicting responses to novel perturbations.

A key assumption in matching methods is the ignorability condition. It posits that, conditional on an adequate set of observed covariates, no unmeasured confounding factors would bias the comparison between control and treated cell populations. Within this framework, given a data set $\mathcal{D} = \{\boldsymbol{X}, \boldsymbol{y}, \boldsymbol{c}\}$, representing $\boldsymbol{M}$ types of perturbation and $\mathbf{G}$ genes, where $\boldsymbol{X} = \{x_1, x_2, ..., x_M\}$ denotes the set of perturbations, $\boldsymbol{y} \in \mathcal{R}^G$ is the post-perturbation gene expression profile, and $\boldsymbol{c}$ is the pre-perturbation (control) gene expression profile, the predicted post-perturbation expression $\hat{\boldsymbol{y}}_{x_i}$ can be modeled as:

$$\hat{\boldsymbol{y}}_{x_i} = c + f(x_i). \tag{1}$$

where $f(x_i)$ is the learned effect under the perturbation $x_i$.

Examples of direct matching models include CellOracle [12], which uses scRNA-seq and scATAC-seq data to infer gene networks for simulating linear perturbation effects. GEARS pioneered predictions for unseen perturbations using gene embeddings and Gene Ontology (GO) based perturbation embeddings. GraphVCI [13] employs counterfactual concepts from causal inference to enhance gene regulatory network learning. sams-VAE [14], uses a sparse additive mechanism shift variational autoencoder to disentangle specific perturbation effects. More recently, single-cell foundation models like scGPT and scFoundation [15] also perform single-cell response prediction as a downstream task.

Despite these advances, robust uncertainty is rarely reported. Some methods, such as GEARS (which uses Monte Carlo dropout [16]), do not fully account for the uncertainty from pair-wise distance to controls. Moreover, the variance of GEARS estimates is not well-calibrated, as it is inconsistent with generalization difficulty and actual prediction accuracy. These limitations underscore the need for more practical and specifically tailored uncertainty quantification frameworks for this task.

### 2.2 Natural Posterior Network for Uncertainty Quantification

The Natural Posterior Network (NatPN) [9] is an evidential deep learning method (Appx.§ E.3) for quantifying uncertainty. Within a single forward process, it estimates epistemic uncertainty (model's confidence) through the distance between the learned posterior and the prior in the latent space, while aleatoric uncertainty (data randomness) is measured via predictive entropy.

NatPN updates beliefs using the Bayesian posterior theory (Appx.§ E.2) for exponential family distributions. Briefly, for a likelihood $\mathbb{P}(\boldsymbol{y} \mid \boldsymbol{\omega})$ and its conjugate prior $\mathbb{Q}(\boldsymbol{\omega} \mid \boldsymbol{\chi}^{\text{prior}}, n^{\text{prior}})$, observing M data points $\{\mathbf{y}_j\}_{j=1}^{M}$ leads to a posterior $\mathbb{Q}(\boldsymbol{\omega} \mid \boldsymbol{\chi}^{\text{post}}, n^{\text{post}})$ with parameters are updated as:

$$\begin{cases} \boldsymbol{\chi}^{\text{post}} = \frac{n^{\text{prior}}\boldsymbol{\chi}^{\text{prior}} + \sum_j^M \boldsymbol{u}(y_j)}{n^{\text{prior}} + M} \\ n^{\text{post}} = n^{\text{prior}} + M \end{cases} . \tag{2}$$

NatPN measures confidence with $n^{\text{post}} - \mathbb{H}[\mathbb{P}(\boldsymbol{y} \mid \boldsymbol{\omega})]$, where $n^{\text{post}}$ (total evidence) captures epistemic uncertainty and entropy represents aleatoric uncertainty. The design of its density estimation architecture has been proven to drive the estimated evidence towards zero under out-of-distribution (OOD) conditions (e.g., far from the training set) (See Appx.§ F.6 for details).

This confidence score shares conceptual similarities with E-distance calculations (Fig. 2(IV)). However, standard NatPN cannot be directly applied to single-cell perturbation prediction. It lacks mechanisms to incorporate pair-wise distances and does not offer a readily available multivariate extension suitable for complex gene expression data. Our work tries to resolve these limitations.

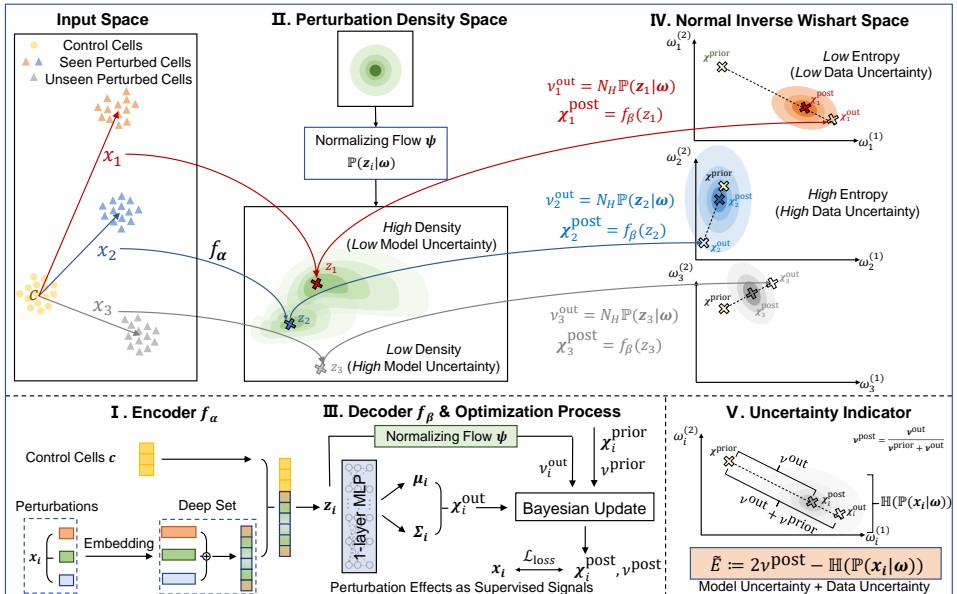

Figure 2: Overview of PRESCRIBE. Each perturbation $X_i$ under condition $c$ is first mapped onto a perturbation density space $z_i$ by the encoder $f_\alpha$. From $z_i$, the decoder $f_\beta$ derives the parameter update $\boldsymbol{\chi}_i^{\text{out}}$ while a normalizing flow $f_\psi$ yields the evidence update $\nu_i^{\text{out}}$. Posterior parameters $\boldsymbol{\omega}_i$ are obtained from a weighted combination of prior $\nu_i^{\text{prior}}$ and updated parameters according to $\nu_i^{\text{out}}$.

## 3 Methods

Our core contribution is the pseudo E-distance, a metric from the model's outputs that unifies epistemic and aleatoric uncertainty. In this section, we first describe the model's probabilistic foundation, then define the pseudo E-distance and its calculation. Finally, we outline the network architecture and the optimization objective employed during training. For reference, detailed notation and hyperparameters are provided in Appx. Tabs. 5 and 6, and the core algorithm in Appx. Algo. 1.

### 3.1 Probabilistic Modeling of Gene Expression

To model both the distribution of gene expression and predictive uncertainty, we adopt a Bayesian approach. For a given perturbation category $x_i$, we assume the gene expression vector $\boldsymbol{y}_i$ follows a multivariate Gaussian distribution. Following related work, we place a Normal-Wishart conjugate prior on the parameters of this Gaussian, which enables analytical Bayesian updates. Thus, a posterior distribution is defined by four parameters $\boldsymbol{\omega}_i = \{\boldsymbol{\mu}_{0x_i} \in \mathbb{R}^N, \kappa_i \in \mathbb{R}^+, \nu_i \geq N, \boldsymbol{L}_i\}$:

$$
\begin{aligned}
\mathbb{P}\left(\boldsymbol{y}_i \mid \boldsymbol{\mu}_i, \boldsymbol{\Lambda}_i\right) &\sim \mathbb{N}\left(\boldsymbol{\mu}_i, \boldsymbol{\Lambda}_i^{-1}\right), & \mathbb{P}(\boldsymbol{\mu}_i \mid \boldsymbol{\Lambda}_i) &= \mathbb{N}\left(\boldsymbol{\mu}_{0x_i}, (\kappa_i \boldsymbol{\Lambda}_i)^{-1}\right), \\
\mathbb{P}(\boldsymbol{\Lambda}_i) &= \mathbb{W}\left(\nu_i, \boldsymbol{\Psi}_i^{-1}\right) = \mathbb{P}(\boldsymbol{\Sigma}_i^{-1}), & \mathbb{P}(\boldsymbol{\Sigma}_i) &= \mathbb{W}^{-1}\left(\nu_i, \boldsymbol{\Psi}_i\right),
\end{aligned}
\tag{3}
$$

where $\boldsymbol{\Sigma}_i = \boldsymbol{\Lambda}_i^{-1}$ is the covariance matrix and $\boldsymbol{\Lambda}_i$ is its corresponding precision matrix. The scale matrix $\boldsymbol{\Psi}_i$ is defined using a lower triangular matrix $\boldsymbol{L}_i$ such that $\boldsymbol{\Psi}_i^{-1} = \nu_i \boldsymbol{L}_i \boldsymbol{L}_i^T$. Here, $\mathbb{N}$, $\mathbb{W}$, and $\mathbb{W}^{-1}$ denote the Normal, Wishart, and Inverse Wishart distributions, respectively. $N$ is the rank of the gene-gene interaction matrix, estimated via Principal Component Analysis (PCA).

The decoder outputs the effect of a perturbation $x_i$ as a set of sufficient statistics and evidence counters:

$$
\begin{cases}
\boldsymbol{\chi}_i^{\text{out}} = \begin{pmatrix} \chi_1^{\text{out}} \\ \chi_2^{\text{out}} \end{pmatrix} = \left( \boldsymbol{\mu}_{0x_i}^{\text{out}} (\boldsymbol{\mu}_{0x_i}^{\text{out}})^T + \dfrac{\boldsymbol{\mu}_{0x_i}^{\text{out}}}{\frac{1}{(\nu_i^{\text{out}})^2}(\boldsymbol{L}_i^{\text{out}})^{-T}(\boldsymbol{L}_i^{\text{out}})^{-1}} \right) \\
n_i^{\text{out}} = \kappa_i^{\text{out}} = 2\nu_i^{\text{out}}.
\end{cases}
\tag{4}
$$

The network's outputs (denoted by the "out" superscript). Since the evidence-related parameters are proportionally linked ($n_i = \kappa_i = 2\nu_i$), we use $\nu$ to refer to them collectively for simplicity.

## 3.2 Pseudo E-distance as a Unified Uncertainty Surrogate

**Definition.** To capture both sources of uncertainty, we define pseudo E-distance (Fig. 2, Panel V) as:

$$\tilde{E} = 2\tilde{\nu}_i^{\text{post}} - \tilde{\mathbb{H}}[\mathbb{P}(\boldsymbol{y}_i \mid \boldsymbol{\omega}_i)]. \tag{5}$$

Here, $\tilde{\cdot}$ denotes normalization. This metric comprises two key terms. The first, $\tilde{\nu}_i^{\text{post}}$, represents the **posterior evidence**, which quantifies the model's epistemic uncertainty. High evidence indicates the prediction is well-supported by training data, while low evidence suggests an out-of-distribution input. The second term, $-\tilde{\mathbb{H}}[\cdot]$, is the negative normalized **entropy** of the predictive distribution, which reflects aleatoric uncertainty or inherent output variability. As shown in Appendix F.5, $\tilde{E}$ provably preserves the rank-ordering of the true E-distance under fixed prior conditions.

**Calculation.** We initialize a base prior using the control cell profile $\boldsymbol{c}$ and fixed hyperparameters ($\kappa_i^{\text{prior}} = 1, \nu_i^{\text{prior}} = 0.5$). The posterior parameters are then obtained by combining this base prior with the model's outputs through the Bayesian update (Eq. 2). From this posterior, the evidence $\nu_i^{\text{post}}$ and the predictive entropy $\mathbb{H}[\cdot]$ are normalized to the range $[N, 2N]$. This operation serves two purposes: it places both components on a comparable scale and it ensures the degrees of freedom of the resulting Student's $t$-distribution remain in a regime that preserves its heavy-tailed properties, which is crucial for capturing sparse regions (low evidence) of the perturbation space.

**Remark 1.** *Our model can distinguish low-confidence, out-of-distribution predictions from high-confidence predictions of perturbation with little to no effect. Although both may predict an outcome similar to the control state, they are separated by their significantly different evidence scores.*

## 3.3 Model Architecture

As illustrated in Fig. 2, our model comprises three core modules: an **encoder** ($f_\alpha$; Panel I) that generates latent representations, a **normalizing flow** ($f_\psi$; Panel II) that estimates evidence, and a **decoder** ($f_\beta$; Panels III–IV) that produces the sufficient statistics for the predictive distribution.

**Encoder $f_\alpha$.** The encoder processes a perturbation $x$ and the cell's basal state $\boldsymbol{c}$ into a latent embedding $\boldsymbol{z} \in \mathbb{R}^D$ that captures functional similarities. Assuming additive effects [17], the encoder comprises two components: $f_{\alpha_1}$ for individual perturbation effects and $f_{\alpha_2}$ for non-linear interactions. For a single perturbation $x_i$, the embedding is calculated as follows:

$$\boldsymbol{z}_i = f_{\alpha_1}(x_i) = f_{\alpha_{13}}(f_{\alpha_{11}}(x_i) + f_{\alpha_{12}}(\boldsymbol{c})). \tag{6}$$

For a set of perturbations $\boldsymbol{x}$, the model aggregates their individual effects via summation. This operation ensures the resulting embedding is invariant to the order of the perturbations:

$$\boldsymbol{z} = f_\alpha(\boldsymbol{x}) = \sum_i f_{\alpha_1}(x_i) + f_{\alpha_2}\left(\sum_i f_{\alpha_1}(x_i)\right). \tag{7}$$

Here, $f_{\alpha_{11}}$ is a linear layer that uses pre-trained gene embeddings, while $f_{\alpha_{12}}$ and $f_{\alpha_2}$ are multilayer perceptrons (MLPs) with LeakyReLU activation. While the encoder design is agnostic to the choice of gene embeddings, we use those from scGPT [10] by default.

**Normalizing Flow $f_\psi$.** The normalizing flow [18] module estimates the density of the training data in the latent space, which directly informs epistemic uncertainty. It takes the latent embedding $\boldsymbol{z}$ as input and outputs the evidence $\nu$. High-density regions (familiar inputs) yield high evidence, while low-density regions (novel inputs) yield low evidence, forcing the prediction to revert toward the pre-defined null state. Specifically, the evidence $\nu_i$ is calculated from the latent embedding $\boldsymbol{z}$, and the posterior evidence $\tilde{\nu}_i^{\text{post}}$ is then updated and normalized as follows:

$$\nu_i = \exp(f_\psi(\mathbf{z}_i) + \ln N_H), \qquad \tilde{\nu}_i^{\text{post}} = \frac{N\nu_i}{\nu_i + \nu^{\text{prior}}} + N \in [N, 2N]. \tag{8}$$

Here, $N_H$ is the total certainty budget, which we set to $N$. The update operation for $\tilde{\nu}_i^{\text{post}}$ scales the posterior evidence to the range $[N, 2N]$ and balances the influence of the base prior (when evidence $\nu_i$ is low) against the data-driven prediction (when evidence $\nu_i$ is high).

**Decoder** $f_\beta$. The decoder is one linear layer that maps the latent embedding $z_i$ to the sufficient statistics $\chi_i^{\text{out}}$. The outputs from the decoder $\chi_i^{\text{out}}$ and the normalizing flow $\nu_i^{\text{out}}$ are then combined with the prior through Bayesian update to form the final posterior distribution and evidence.

### 3.4 Optimization Objective

PRESCRIBE is trained by minimizing a composite loss function $\mathcal{L}$ designed to encourage both accurate predictions and meaningful perturbation density estimates:

$$
\mathcal{L} = - \underbrace{\mathbb{E}_{\boldsymbol{\omega}_i \sim \mathbb{W}_i^{-1,\text{post}}} \left[ \ln \mathbb{P}\left( \boldsymbol{y}_i \mid \boldsymbol{\omega}_i \right) \right]}_{\mathcal{L}_1} - \lambda_1 \| \boldsymbol{y}_i - \boldsymbol{\mu}_{0x_i}^{\text{post}} \|_1 \underbrace{\mathbb{H}\left[ \mathbb{W}_i^{-1,\text{post}} \right]}_{\mathcal{L}_2}
$$

$$
- \lambda_2 \underbrace{\frac{1}{B} \sum_{i=1}^{B} \ln \left( \frac{e^{\tilde{E}_{(\text{sorted}, x_i)}}}{\sum_{j=1}^{B} e^{\tilde{E}_{(\text{sorted}, x_j)}}} \right)}_{\mathcal{L}_3} - \lambda_3 \| \boldsymbol{y}_i - \boldsymbol{\mu}_{0x_i}^{\text{post}} \|_1 \underbrace{\ln \left( \frac{N}{2N - \nu_i} - 1 \right)}_{\mathcal{L}_4}. \tag{9}
$$

Specifically, $\mathcal{L}$ consists of a primary objective for prediction accuracy ($\mathcal{L}_1$) and three auxiliary terms, weighted by hyperparameters $\lambda_1, \lambda_2, \lambda_3$ (see § 4.9 for grid search details):

**Expected Log-Likelihood ($\mathcal{L}_1$).** This term maximizes the likelihood of the observed data under the posterior predictive distribution, driving accurate predictions. The details can be found in Appx. F.1.

**Entropy Regularization ($\mathcal{L}_2$).** This term, weighted by prediction error, acts as a prior that favors uninformative distributions with high entropy. The details can be found in Appx. F.2.

**E-distance Ranking Loss ($\mathcal{L}_3$).** To supervise the model's uncertainty estimates, we introduce a ranking loss based on ListMLE [19]. This loss enforces consistency between the predicted and reference rankings within each training batch. Specifically, the predicted pseudo E-distances, $\tilde{E}$, are reordered according to the descending order of their corresponding reference E-distances, $E$:

$$
\tilde{E}_{\text{sorted}} = \tilde{E}[\text{argsort}(E, \text{descending})]. \tag{10}
$$

The term $\tilde{E}_{(\text{sorted}, x_i)}$ refers to the $i$-th element of this sorted list. This objective encourages the model's predicted ranking of uncertainties to match the reference ranking.

**Uncertainty Regularization Loss ($\mathcal{L}_4$).** This term addresses the problem of vanishing gradients in low-evidence regions. The details can be found in Appx. F.3.

## 4 Results

This section details PRESCRIBE's empirical evaluation. The experiments were designed to: (i) assess the quality of its uncertainty (§ 4.3- 4.6)). (ii) demonstrate its utility in prediction accuracy (§ 4.7), and (iii) explore contributions of its core components and initialization strategies (§ 4.8- 4.9).

### 4.1 Experimental Setup

**Datasets.** We evaluated PRESCRIBE on three widely recognized benchmark datasets: Norman [4], Replogle2022_Rep1, and Replogle2022_K562 [5]. (Details are provided in Appx.§ G.)

**Comparison Baselines.** To evaluate PRESCRIBE, we conducted a comprehensive benchmark against several existing methods. The baseline methods were divided into two main groups: recent methods without uncertainty estimation (**AverageKnown**, **Linear**, **Linear scGPT** [20], **CellOracle** [12], **samsVAE** [14], **GraphVCI** [13], **scFoundation** [15], and **scGPT** [10]) and those with variance-based uncertainty estimation (**GEARS** [1] and **GEARS-Drop**). The evaluation also encompassed several adaptations and ablations of our approach and related models: **GEARS-ens**, an ensemble of five GEARS-Drop models; **Ours-MLPs**, a variant that employed the our model's encoder/decoder architecture but specifically used Multi-Layer Perceptrons (MLPs) to regress E-distance from the latent embedding; **Ours-Null**, representing our model's untrained state; **Ours-Ens**,

an ensemble of five models using the our model's encoder and an MC dropout [16] decoder; and **Ours-NOINFO**, a variant that removed prior information by using zero vectors for the prior mean and covariance. The complete settings of these baselines can be found in Appx. §H.

## 4.2 Evaluation Metrics

Our evaluation employed two categories of metrics, each serving a different role.

**Prediction Accuracy.** These metrics are assessed by measuring the Pearson Correlation Coefficient ($r_{\text{pred,truth}}$) and directional accuracy ($\text{ACC}_{\text{pred,truth}}$) between predicted and true gene expression. We also calculated these metrics on the top 20 differentially expressed genes (DEGs), denoted as $r_{\text{pred,truth}}^{\text{DEG}}$ and $\text{ACC}_{\text{pred,truth}}^{\text{DEG}}$. All accuracy evaluations were performed on log-fold change values.

**Calibration Quality.** These metrics assess if a model's confidence scores are meaningful indicators of its predictive performance. We measure this using several metrics: **(1)** the Pearson ($r_{\text{perf,conf}}$) and Spearman's ($r_{\text{perf,conf}}^{s}$) correlation between confidence scores and actual performance (as measured by $r_{\text{pred,truth}}$); **(2)** a percentile-based classification accuracy ($\text{ACC}_{\text{perf,conf}}$) to determine if low-confidence predictions are less accurate; and **(3)** the Expected Calibration Error (ECE) [21], denoted as $\epsilon_{\text{perf,conf}}$.

## 4.3 Estimation of E-distance

Our model is designed to efficiently estimate E-distance without requiring post-perturbation profiles. We therefore validated the practical effectiveness of our estimated "pseudo E-distance" ($\hat{E}$) in two dimensions. First, across all datasets, $\hat{E}$ exhibited a consistent positive correlation with a reference E-distance ($E$), which was computed using post-perturbation profiles (Tab. 1(a)). Second, this positive correlation strengthened significantly as the number of samples ($N$) used for computing the reference E-distance ($E$) increased (Tab. 1(b)). Given that $E$ calculated with a larger $N$ more accurately reflects the true underlying E-distance, this observed trend suggests that $\hat{E}$ can serve as an effective, and potentially asymptotic, approximation of the true E-distance's ranking.

Table 1: Comparison of pseudo $\hat{E}$ with reference E-distance ($E$): (a) across various datasets; (b) across increasing sample size ($N$) used for reference $E$ calculation on the Norman dataset.

| | (a) Cross dataset performance | | | | | | | | |
| | **Norman.** | | | **Rep1.** | | | **K562.** | | |
| Method | $r_{\text{E,conf}}^{s}$ ↑ | $\text{ACC}_{\text{E,conf}}$ ↑ | $r_{\text{E,conf}}$ ↑ | $r_{\text{E,conf}}^{s}$ ↑ | $\text{ACC}_{\text{E,conf}}$ ↑ | $r_{\text{E,conf}}$ ↑ | $r_{\text{E,conf}}^{s}$ ↑ | $\text{ACC}_{\text{E,conf}}$ ↑ | $r_{\text{E,conf}}$ ↑ |
|---|---|---|---|---|---|---|---|---|---|
| Ours-Null | 0.76 | 16.13 | -3.28 | -3.61 | 19.10 | -2.34 | -9.58 | 17.65 | -10.96 |
| Ours-MLP | 27.49 | 25.03 | 29.56 | -45.64 | 22.81 | -50.59 | -3.61 | 18.89 | -2.34 |
| Ours | 35.56 | 25.81 | 33.84 | 12.18 | 23.53 | 21.80 | 24.74 | 31.58 | 16.86 |

| | (b) Cross sample size $N$ performance | | | | | | | | |
| | **Ours** | | | **Ours-MLPs** | | | **Ours-Null** | | |
| $N$ | $r_{\text{E,conf}}^{s}$ ↑ | $\text{ACC}_{\text{E,conf}}$ ↑ | $r_{\text{E,conf}}$ ↑ | $r_{\text{E,conf}}^{s}$ ↑ | $\text{ACC}_{\text{E,conf}}$ ↑ | $r_{\text{E,conf}}$ ↑ | $r_{\text{E,conf}}^{s}$ ↑ | $\text{ACC}_{\text{E,conf}}$ ↑ | $r_{\text{E,conf}}$ ↑ |
|---|---|---|---|---|---|---|---|---|---|
| 54 | 35.56 | 25.81 | 33.84 | 29.03 | 27.49 | 29.56 | 16.13 | 0.76 | -3.28 |
| 200 | 41.37 | 33.33 | 45.44 | 33.33 | 32.71 | 31.52 | 7.69 | -14.68 | -9.42 |
| 500 | 80.00 | 50.00 | 67.55 | 50.00 | 20.00 | 24.22 | NaN | 33.33 | NaN |

## 4.4 Confidence Scaling with Generalization Difficulty

We utilized the Norman combination perturbation dataset to assess our model's performance under varying levels of generalization difficulty. We simulated increasing difficulty by varying the number of unseen perturbations in combinations of two (0, 1, or 2 unseen). To remove the confounding variables from E-distance, we controlled for the average E-distance across these scenarios, approximately 55. As illustrated in Fig. 3(a), our model significantly reduced its confidence scores as the generalization difficulty increased. In contrast, others show a minorly decrease or inverse trend. This phenomenon demonstrates our model's ability to be more aware of epistemic uncertainty than other baselines.

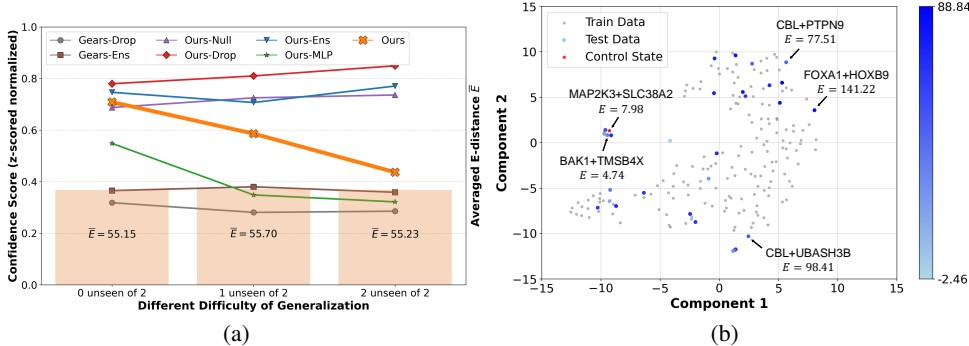

Figure 3: (a) The line plot depicts confidence scores from different models plotted against generalization difficulty. An accompanying bar plot displays the average E-distance for corresponding levels of generalization difficulty. (b) t-SNE visualization of the Normal-Inverse Wishart space.

Table 2: Perturbation Prediction Performance Comparison (in %)

| Models | Norman. | | | | Rep1. | | | | K562. | | | |
|---|---|---|---|---|---|---|---|---|---|---|---|---|
| | $r_{\text{pred,truth}}$ ↑ | $r^{\text{DEG}}_{\text{pred,truth}}$ ↑ | $\text{ACC}_{\text{pred,truth}}$ ↑ | $\text{ACC}^{\text{DEG}}_{\text{pred,truth}}$ ↑ | $r_{\text{pred,truth}}$ ↑ | $r^{\text{DEG}}_{\text{pred,truth}}$ ↑ | $\text{ACC}_{\text{pred,truth}}$ ↑ | $\text{ACC}^{\text{DEG}}_{\text{pred,truth}}$ ↑ | $r_{\text{pred,truth}}$ ↑ | $r^{\text{DEG}}_{\text{pred,truth}}$ ↑ | $\text{ACC}_{\text{pred,truth}}$ ↑ | $\text{ACC}^{\text{DEG}}_{\text{pred,truth}}$ ↑ |
| AverageKnown | 39.64 | 58.98 | 27.23 | 61.94 | 54.53 | 57.89 | 53.66 | 32.33 | 36.86 | 46.11 | 59.18 | 56.14 |
| Linear | 37.68 | 55.54 | 26.87 | 61.94 | 38.01 | 40.70 | 47.65 | 30.15 | 25.70 | 32.42 | 52.54 | 52.36 |
| Linear-scGPT | 39.20 | 58.66 | 27.23 | 61.94 | 50.09 | 53.95 | 49.69 | 31.31 | 33.86 | 42.97 | 54.79 | 54.37 |
| CellOracle | 9.80 | 12.48 | 19.20 | 16.35 | 39.91 | 7.40 | 37.55 | 23.70 | 4.44 | 5.89 | 41.41 | 41.15 |
| samsVAE | 12.48 | 32.05 | 37.42 | 49.63 | 12.59 | 36.45 | 33.04 | 25.08 | 8.51 | 29.03 | 36.44 | 43.55 |
| GraphVCI | 12.02 | 30.66 | 27.95 | 33.95 | 14.39 | 36.30 | 41.41 | 25.08 | 9.73 | 28.91 | 45.66 | 43.55 |
| scFoundation | 60.79 | 65.65 | 35.66 | 62.26 | 47.60 | 59.46 | 53.38 | 43.96 | 25.15 | 47.30 | 57.11 | 57.32 |
| scGPT | 61.48 | 65.87 | 61.96 | 74.43 | 50.32 | 65.54 | 61.72 | 67.07 | 32.72 | 43.15 | 57.44 | 57.32 |
| GEARS | 45.30 | 63.19 | 29.09 | 69.06 | 48.18 | 53.59 | 51.08 | 32.33 | 32.57 | 42.68 | 56.33 | 56.14 |
| GEARS-Drop | 44.96 | 60.38 | 29.92 | 68.28 | 46.49 | 51.05 | 52.25 | 37.08 | 31.26 | 42.88 | 56.34 | 57.67 |
| GEARS-Drop-5% | 45.05 | 59.61 | 29.72 | 66.83 | 47.01 | 51.27 | 52.25 | 36.79 | 31.66 | 43.54 | 56.28 | 57.44 |
| GEARS-Drop-10% | 49.72 | 65.23 | 30.57 | 70.18 | 45.34 | 48.68 | 52.09 | 36.93 | 31.94 | 43.65 | 56.48 | 58.22 |
| GEARS-Ens | 45.94 | 62.44 | 29.21 | 69.38 | 47.87 | 49.92 | 51.91 | 34.30 | 30.58 | 42.99 | 56.22 | 56.36 |
| GEARS-Ens-5% | 45.95 | 61.85 | 29.01 | 68.00 | 48.32 | 49.72 | 51.91 | 33.94 | 30.99 | 43.60 | 56.16 | 56.12 |
| GEARS-Ens-10% | 50.91 | 67.42 | 29.86 | 71.43 | 48.05 | 50.07 | 51.99 | 34.06 | 31.29 | 43.84 | 56.42 | 56.91 |
| PRESCRIBE-Null | 14.40 | 43.84 | 51.19 | 65.97 | 8.50 | 16.95 | 51.83 | 55.89 | 10.96 | 21.80 | 52.38 | 56.30 |
| PRESCRIBE | 58.38 | 64.44 | 63.24 | 74.68 | 59.18 | 65.50 | 67.36 | 79.81 | 36.20 | 44.36 | 60.27 | 69.69 |
| PRESCRIBE-5% | 61.58 | 66.36 | 64.08 | 75.69 | 60.20 | 66.07 | 67.76 | 79.94 | 38.28 | 46.63 | 60.99 | 71.15 |
| PRESCRIBE-10% | 64.32 | 68.61 | 64.73 | 75.93 | 60.28 | 66.13 | 67.89 | 80.03 | 38.58 | 47.52 | 61.04 | 71.21 |

## 4.5 Qualitative Visualization

We visualized the Normal Inverse Wishart latent space learned by our model using t-SNE [22] dimensionality reduction on the Norman dataset (Fig. 3(b)). This visualization represents control (prior), training, and test perturbations by red, gray, and blue, respectively. The intensity of the blue color corresponds to the $r_{\text{pred,truth}}$ prediction accuracy, with darker shades indicating higher quality predictions. The visualization aligns with expectations: training data points and large $E$ values are generally located further from the control prior; the model assigns them higher confidence, reflecting their dominance over the prior, and indeed, most of them exhibit high prediction accuracy $r_{\text{pred,truth}}$.

## 4.6 Accuracy-Based Calibration Analysis

We examined accuracy-based calibration curves for a more granular understanding of our model's practical utility. Furthermore, we computed the Spearman correlation ($r^s_{\text{perf,conf}}$) between prediction accuracy and the confidence scores produced by our model and baseline methods (for GEARS-Drop and GEARS-Ens, inverse values were used as their confidence scores represent variance). From Fig. 4, our model achieved the highest $r^s_{\text{perf,conf}}$ across all datasets, and the visualizations of the calibration curves demonstrated that our model exhibits a consistent trend where confidence increased monotonically with accuracy, indicating its confidence score is well-calibrated overall. The expected calibration error ($\epsilon_{\text{perf,conf}}$) results are presented in the supplementary materials (Appx. I).

## 4.7 Prediction Accuracy with Uncertainty-Guided Filtering

A practical use of reliable uncertainty estimation is identifying and potentially excluding low-confidence predictions, thereby improving overall performance. We investigated this by comparing our model's accuracy after filtering out the 5% and 10% least confident predictions (Ours 5% and Ours 10%) against baseline models (Tab. 2). Ours-10% significantly outperformed all baselines, while Ours-5% also showed strong results. It is important to note that modeling distributions (as our model

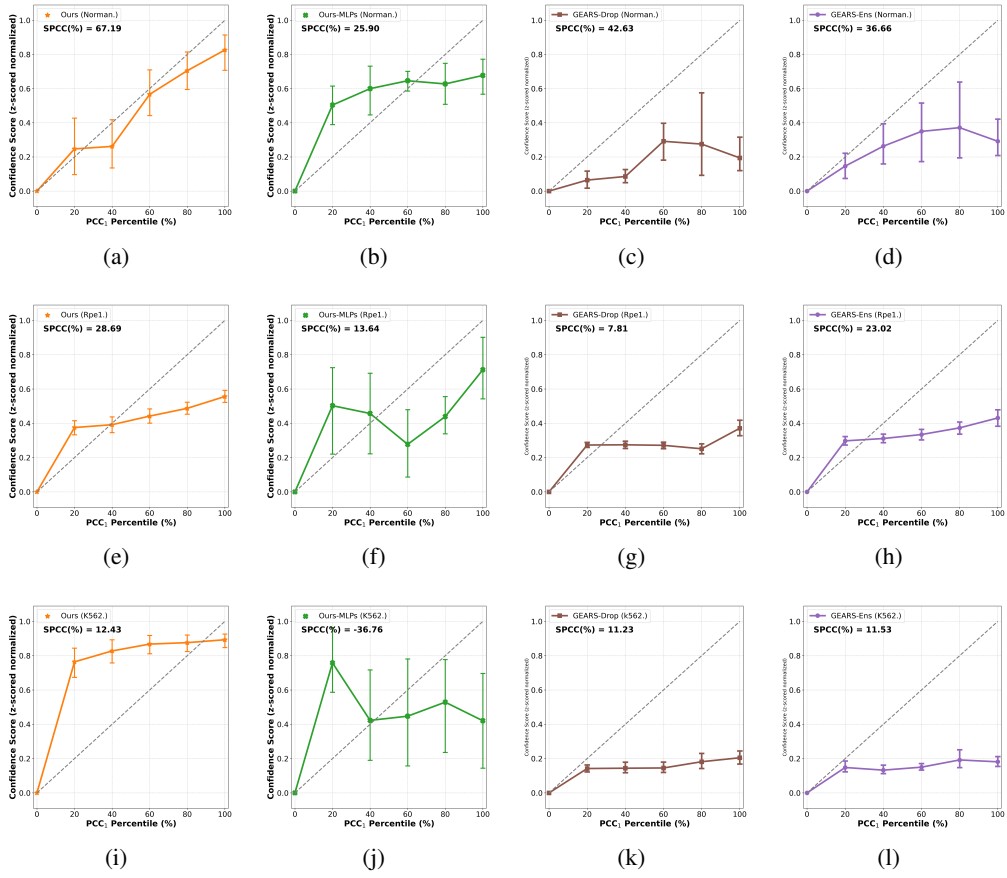

Figure 4: Calibration curves relating model-estimated confidence scores to prediction accuracy ($r_{\text{pred,truth}}$). Each row corresponds to a dataset, while columns and colors distinguish different models (in order: Ours, Ours-MLPs, GEARS-Drop, and GEARS-Ens). Text in each subplot indicates the Spearman Correlation Coefficient ($r^s_{\text{perf,conf}}$) between $r_{\text{pred,truth}}$ and confidence scores.

does) is inherently more complex than direct value prediction. Thus, models like GEARS-Drop and Ours, which learn an additional function for confidence evaluation, did not exhibit standout accuracy initially. However, only our model demonstrated stable, significant, consistent improvements across metrics after different degrees of filtering unreliable predictions.

### 4.8 Ablation Studies on Filtering and Key Model Components

We conducted several ablation studies further to understand the sources of our model's performance.

Table 3: Ablation Studies on Filtering (in %)

| Model | $r_{\text{pred,truth}} \uparrow$ | $\text{ACC}_{\text{pred,truth}} \uparrow$ |
|---|---|---|
| Ours | 58.38 | 63.24 |
| Random Filtering-5% | 53.35±3.16 | 60.56±0.97 |
| Random Filtering-10% | 58.28±1.59 | 57.49±2.25 |

Table 4: Ablation Studies on Modules (in %)

| Model | $r_{\text{pred,truth}} \uparrow$ | $\text{ACC}_{\text{pred,truth}} \uparrow$ | $r^s_{\text{perf,conf}} \uparrow$ |
|---|---|---|---|
| Ours-NOINFO | 43.25 | 57.80 | 36.65 |
| w/o $\mathcal{L}_3$ | 55.61 | 63.61 | 27.74 |
| w/o $\mathcal{L}_4$ | 22.41 | 54.73 | 29.83 |

**Impact of Filtering.** We performed random filtering (10 repetitions) to assess whether the observed performance gains were merely an artifact of data removal. As shown in Tab. 3, randomly filtering 5% of predictions significantly decreased accuracy. Filtering 10% randomly brought accuracy back to approximately the initial unfiltered level, but did not yield significant improvements. This confirms that the performance gains of our model are not attributable to the filtering operation.

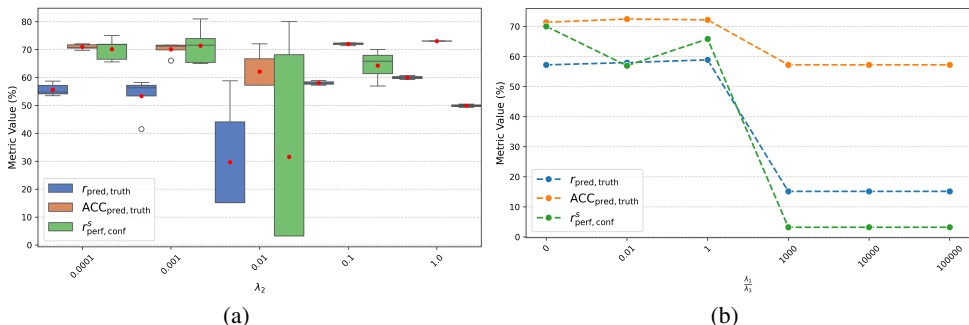

Figure 5: Hyperparameter Search on Norman Dataset. (a) Box plots illustrating the relationship between $\lambda_2$ and the distributions of $r_{\text{pred,truth}}$, $\text{ACC}_{\text{pred,truth}}$, and $r^s_{\text{perf,conf}}$. (b) Line plots illustrating the relationship between the ratio $\lambda_1/\lambda_3$ and metrics $r_{\text{pred,truth}}$, $\text{ACC}_{\text{pred,truth}}$, and $r^s_{\text{perf,conf}}$, when $\lambda_2 = 0.1$.

**Impact of Key Components.** As demonstrated in Tab. 4, evaluating specific architectural and loss components revealed their distinct contributions. The Ours-NOINFO variant, despite comparable prediction performance to our model, showed significantly lower $r^s_{\text{perf,conf}}$ values, underscoring the importance of informative prior settings for well-calibrated uncertainty. The loss term $\mathcal{L}_3$, which supervises learning from E-distance, proved critical; its removal led to a slight $\text{ACC}_{\text{pred,truth}}$ increase, potentially due to the adversarial nature of optimizing accuracy versus evidential uncertainty, but at the cost of substantially reduced uncertainty calibration. Finally, ablating the loss term $\mathcal{L}_4$ resulted in a suboptimal model due to zero gradients, as mentioned in § 3.4.

## 4.9 Hyperparameter Search Strategy

We developed an efficient hyperparameter tuning protocol for applying our model to new datasets.

**Selection Pipeline.** The selection protocol can be summarized in the following steps:

- Phase 1 - $\lambda_2$ Optimization: Conduct a grid or random search for $\lambda_2$ (the weight for $\mathcal{L}_3$).

- Phase 2 - $\lambda_1/\lambda_3$ Ratio Tuning: Fix $\lambda_3$ (the weight for $\mathcal{L}_4$) to $10^{-5}$ and perform a focused search for $\lambda_1$ (the weight for $\mathcal{L}_2$). This effectively tunes the ratio $\lambda_1/\lambda_3$.

**Rationale and Example.** Our efficient two-phase tuning pipeline reduces trials from $216$ ($6 \times 6 \times 6$) to $42$ ($6 \times 6 + 6$) by fixing $\lambda_3$ in the second phase. This is justified by the adversarial relationship between $\mathcal{L}_2$ (weighted by $\lambda_1$, promoting prior adherence) and $\mathcal{L}_4$ (weighted by $\lambda_3$, encouraging evidence accumulation). Thus, we fix $\lambda_3 = 10^{-5}$ and tune $\lambda_1$. For example, on the Norman dataset, an initial random search determined $\lambda_2 = 0.1$ (Fig. 5(a)). Tuning the $\lambda_1/\lambda_3$ ratio (Fig. 5(b)) then revealed an inverse relationship between confidence calibration ($r^s_{\text{perf,conf}}$) and prediction performance ($r_{\text{pred,truth}}$, $\text{ACC}_{\text{pred,truth}}$), leading us to select $\lambda_1/\lambda_3 = 0.01$ (so $\lambda_1 = 10^{-7}$ for $\lambda_3 = 10^{-5}$).

## 5 Conclusion and Discussion

**Conclusion.** We introduce PRESCRIBE, a novel uncertainty-aware framework for single-cell response prediction. By employing a multivariate Natural Posterior Network to estimate the pseudo E-distance for each perturbation category, our model delivers well-calibrated, instance-level uncertainty. This metric acts as a unified surrogate for both aleatoric (data) and epistemic (model) uncertainty, and its application demonstrates practical utility by improving overall predictive accuracy.

**Future Work.** Recent research [23] highlights the critical impact of the reference state (often the unperturbed cell state) in this task. Therefore, future work will leverage PRESCRIBE's configurable "null" state to explore how the choice of this reference influences predicted perturbation responses. The goal is to guide the optimal selection of a reference state, enabling the model to learn the specific effects of perturbations more easily along the pronounced direction of change.

# 6 Acknowledgment

This work is supported by STI2030-Major Projects 2022ZD0212400, the National Natural Science Foundation of China Project (No. 623B2086), TeleAI, STCSM grant 24510714300 and 20DZ2254400, GuangDong Basic and Applied Basic Research Foundation grant 2023B1515120006, and SJTU Science-Medicine interdisciplinary grant 24X010301456.

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

# Appendix Overall

# A  Notation Table

Table 5: Notation and Descriptions for Key Variables in PRESCRIBE

| Notation | Description |
|---|---|
| $\mu$ | Mean value |
| $\Sigma$ | Covariance matrix |
| $M$ | Number of perturbation types |
| $X$ | Genetic perturbation |
| $y$ | Post-perturbed transcriptomics expressions |
| $c$ | Pre-perturbed transcriptomics expressions |
| $\mathbb{P}$ | Exponential family distribution |
| $\mathbb{Q}$ | Conjugate prior |
| $\mathbb{N}$ | Normal distribution |
| $\sigma$ | Variance |
| $\mathbb{W}$ | Wishart distribution |
| $\mathbb{W}^{-1}$ | Inverse Wishart distribution |
| $\Lambda$ | Precision matrix |
| $N$ | Rank of gene-gene interaction matrix |
| $d$ | Hidden dim |
| $D$ | Latent dim |
| $S$ | Token count in scGPT embedding |
| $\mu_0, \nu, \kappa, L$ | Parameters in NIW†; $L$: lower triangular matrix |
| $\Psi$ | NIW parameter: $\Psi = \frac{1}{\nu} L^{-\top} L^{-1}$ |
| $\omega$ | Simplified NIW parameters: $\omega = \{\mu_0, \kappa, \nu, L\}$ |
| $n$ | Evidence (data observations) |
| $\chi$ | Sufficient statistics for NIW |
| $\hat{E}$ | Pseudo E-distance |
| $E$ | E-distance (energy distance) |
| $B$ | Batch size |
| $\mathbb{H}$ | Entropy |
| $\psi$ | Digamma function |
| $\Gamma$ | Gamma function |
| $\theta_\alpha$ | Encoder learnable parameters |
| $\theta_\beta$ | Decoder learnable parameters |
| $\theta_\psi$ | Flow model learnable parameters |
| $\mathcal{L}_1$ | Expected log-likelihood |
| $\mathcal{L}_2$ | Entropy regularization |
| $\mathcal{L}_3$ | E-distance supervised loss |
| $\mathcal{L}_4$ | Uncertainty-regularized loss |
| $\lambda_{1-3}$ | Weights for $\mathcal{L}_2, \mathcal{L}_3, \mathcal{L}_4$ |
| $r_{\text{pred,truth}}$ | Pearson correlation (predicted vs. true log fold change) |
| $\text{ACC}_{\text{pred,truth}}$ | Directional accuracy (predicted vs. true log fold change) |
| $r_{\text{perf,conf}}$ | Pearson correlation between $\hat{E}$ and $x*$ |
| $\text{ACC}_2$ | Quintile classification accuracy ($\hat{E}$ vs. $x*$) |
| $r^s_{\text{perf,conf}}$ | Spearman correlation ($\hat{E}$ vs. $x*$) |

†NIW = Normal Inverse Wishart distribution; $*x$ defaults to $\text{PCC}_1$ unless specified.

# B Default Settings

The PRESCRIBE framework was trained on a single NVIDIA RTX 4090 GPU. We used the Adam [24] optimizer with an initial learning rate of $10^{-4}$ and set the weight decay value as $10^{-5}$. The training was carried out for a maximum of $50$ epochs using batch sizes of $4096$ samples, with early stopping triggered if $\mathcal{L}_1$ of the validation set did not show improvement for three consecutive epochs. A dynamic learning rate scheduler "plateau" reduced the learning rate by 1% when the relative $\mathcal{L}_1$ validation improvement remained below 0.01% for two consecutive epochs. The implementation leverages PYTORCH LIGHTNING [25] 2.4.0's modular architecture.

Table 6: Experiment Configurations.

|  | Hyperparameter | Value |
|---|---|---|
| **Model Settings** | Flow layers | 10 |
|  | Flow size | 0.774231384 |
|  | Flow n hidden | 2 |
|  | Bound | 30 |
|  | $D$ | 64 |
|  | $N$ | 10 |
| **Training Details** | Warm up epochs | 5 |
|  | Warm up learning rate | 0.001 |
|  | Optimizer | Adam |
|  | Scheduler | Plateau |
|  | Scheduler change step | 2 |
|  | Scheduler reduce rate | 0.99 |
|  | Patience | 3 |
|  | Learning rate | 1e-4 |
|  | Total epochs | 50 |
|  | Weight decay | 1e-5 |
|  | $B$ | 4096 |
|  | Gradient Accumulation | 4 |
|  | $\lambda_1$ | 1e-7 |
|  | $\lambda_2$ | 0.1 |
|  | $\lambda_3$ | 1e-5 |

## C   Algorithm

The optimization process of PRESCRIBE can be summarized in the following algorithm 1

---
**Algorithm 1** Training Process of PRESCRIBE
---
1: **Input:**
2:      Perturbation types: $\boldsymbol{X}$;
3:      Condition information: $\boldsymbol{c}, \boldsymbol{\chi}^{prior}$;
4:      Prior evidence: $\nu^{prior}$;
5:      Post-perturbed transcriptomics expressions: $\boldsymbol{y}_i$;
6:      E-distance of training samples: $E_i$;
7: **Initialize:**
8:      $\boldsymbol{\theta} = \{\theta_\alpha, \theta_\psi, \theta_\beta\} \leftarrow$ initialize network parameters;
9: **repeat**
10:        $x_i \leftarrow$ random mini-batch from $\boldsymbol{X}$;
11:        $z_i \leftarrow f_\alpha(x_i, \boldsymbol{c})$; // Encoder;
12:        $n_i \leftarrow f_\phi(z_i)$; // Flow;
13:        $\boldsymbol{\chi}_i = \{\chi_1, \chi_2\} \leftarrow f_\beta(z_i)$; // Decoder;
14:        // Bayesian Posterior Update;
15:        $\boldsymbol{\chi}_i \leftarrow$ compute through Eq. 4;
16:        $\nu_i^{\text{post}}, \nu_i \leftarrow$ compute through Eq. 8;
17:        $\kappa_i^{\text{post}} \leftarrow 2 \cdot \nu_i^{\text{post}}$;
18:        $\boldsymbol{\chi}_i^{\text{post}} \leftarrow \frac{(\nu^{\text{prior}}\boldsymbol{\chi}^{\text{prior}} + \nu_i \boldsymbol{\chi}_i)}{\nu_i + \nu^{\text{prior}}}$;
19:        $\boldsymbol{\mu}_{0x_i}^{\text{post}} \leftarrow \chi_1^{\text{post}}$;
20:        $\boldsymbol{L}^{\text{post}} \leftarrow$ Cholesky$((\chi_2^{\text{post}} - (\chi_1^{\text{post}})^2) \times (\nu_i^{\text{post}})^2) \times \nu_i^{\text{post}}$;
21:        $\mathcal{L} \leftarrow$ compute through Eq. 9;
22:        // Update parameters according to gradients;
23:        $\boldsymbol{\theta} \leftarrow \boldsymbol{\theta} - \nabla_{\boldsymbol{\theta}}\mathcal{L}$;
24: **until** deadline reached
---

## D   Computational Complexity Analysis

The computational complexity of our model is characterized by three primary components: the Encoder, the Flow, and the Decoder. The complexities for each component are detailed below:

1. **Encoder**: The overall computational complexity for the Encoder is $\mathcal{O}(d \cdot (S + G + d + D))$. This is derived from the complexities of its constituent functions:
    - $f_{11}$: $\mathcal{O}(S \cdot d + d)$
    - $f_{12}$: $\mathcal{O}(G \cdot d + d)$
    - $f_{13}$: $\mathcal{O}(d \cdot d + d)$
    - $f_2$: $\mathcal{O}(d \cdot d + d)$
    - $f_o$: $\mathcal{O}(d \cdot D + D)$

2. **Flow**: The complexity associated with the Flow component is $\mathcal{O}(D)$.

3. **Decoder**: For the Decoder, which takes an input of dimension $D$ and produces an output of dimension $(N \cdot (N + 1)/2 + N)$, the computational complexity is $\mathcal{O}(D \cdot N^2)$.

## E   Preliminaries

### E.1   E-Distance

E-distance [26], a statistical metric for assessing high-dimensional distributions, offers measures of the signal-to-noise ratio within a dataset. It is popularly referred to as energy distance, originally derived from the concept of gravitational energy in physics. In essence, E-distance determines

whether two distributions are distinct by comparing the distance between them – pairwise distance– to the variability within each distribution–self-distance". The illustration can be referred to in Fig.2(IV). Applying E-distance in single-cell perturbation prediction task was introduced by scPerturb [8], which utilized E-distance to differentiate between strong and weak perturbations. For two given single-cell statuses, represented as $X$ and $Y$, the E-distance calculation is defined as follows:

$$E(X,Y) := 2\delta_{XY} - \sigma_X - \sigma_Y, \tag{11}$$

where $\sigma_X$ and $\delta_{XY}$ are the self-distance within and pairwise distance between distributions:

$$\sigma_X = \frac{1}{N(N-1)} \sum_{i=1}^{N} \sum_{j=1}^{N} \|x_i - x_j\|, \tag{12}$$

$$\delta_{XY} = \frac{1}{NM} \sum_{i=1}^{M} \sum_{j=1}^{N} \|x_i - y_j\|. \tag{13}$$

Typically, scPerturb sets the control status as $X$, aligning with most deep learning models' assumptions. Our observation (Fig. 1(b)) suggests an apparent positive correlation between E-distance and predictive quality. However, since E-distance needs to access data, it poses challenges for predicting unseen perturbations, which is out-of-sample data.

## E.2 Bayesian Posterior Update

In the context of distributions from the exponential family, we can express the conjugate prior and the posterior distribution after observing $M$ target observations as follows:

$$\mathbb{P}(\boldsymbol{y} \mid \boldsymbol{\omega}) = h(\boldsymbol{y}) \exp\left(\boldsymbol{\theta}^T \boldsymbol{u}(\boldsymbol{y}) - A(\boldsymbol{\omega})\right), \tag{14}$$

$$\mathbb{Q}(\boldsymbol{\omega} \mid \boldsymbol{\chi}, n) = \eta(\boldsymbol{\chi}, n) \exp\left(n\boldsymbol{\omega}^T \boldsymbol{\chi} - nA(\boldsymbol{\omega})\right), \tag{15}$$

$$\mathbb{Q}\left(\boldsymbol{\omega} \mid \boldsymbol{\chi}^{\text{post}}, n^{\text{post}}\right) \propto \exp\left(n^{\text{post}} \boldsymbol{\omega}^T \boldsymbol{\chi}^{\text{post}} - n^{\text{post}} A(\boldsymbol{\omega})\right). \tag{16}$$

The parameters $\omega$, $\chi$, and $n$ correspond to the target distribution, prior distribution, and evidence, respectively. The posterior parameters can be updated as follows:

$$\begin{cases} \chi^{\text{post}} = \frac{n^{\text{prior}} \chi^{\text{prior}} + \sum_{j}^{M} \boldsymbol{u}(y_j)}{n^{\text{prior}} + M} \\ n^{\text{post}} = n^{\text{prior}} + M \end{cases}. \tag{17}$$

The method nature posterior network [9] has leveraged this framework to predict Bayesian uncertainty. Moreover, it employs normalizing flows [18] to develop a structural latent space, inferring that regions distant from the training data should exhibit higher uncertainty. However, this approach has yet to be expanded to the multivariate Gaussian distribution, which needs to be theoretically complemented in corresponding derivation to predict single-cell perturbation response better.

## E.3 Deep Evidential Regression.

Evidential learning provides a promising solution to the challenge of overconfidence in deep learning models. This approach is more efficient than ensemble methods [27] or Monte Carlo approximations, as it learns high-order prior distributions by gathering evidence from training samples in a single forward pass. Currently, the research on multivariate regression within this framework is limited, with only one study [28] focusing on the parameters of a Normal Inverse-Wishart (NIW) distribution, which serves as a conjugate prior. In this setting, the target $\boldsymbol{y} \in \mathcal{R}^N$ is considered to be independently and identically distributed (i.i.d.) from a multivariate Gaussian distribution with an unknown mean $\boldsymbol{\mu}$ and covariance matrix $\boldsymbol{\Sigma}$. The model aims to predict not only the mean $\mathbb{E}[\boldsymbol{\mu}]$, but also the aleatoric uncertainty $\mathbb{E}[\boldsymbol{\Sigma}]$ and epistemic uncertainty $\text{var}[\boldsymbol{\mu}]$. These predictions can be expressed as follows:

$$\mathbb{E}[\boldsymbol{\mu}] = \boldsymbol{\mu}_0, \quad \mathbb{E}[\boldsymbol{\Sigma}] \propto \frac{\nu}{\nu - N - 1} \boldsymbol{L} \boldsymbol{L}^\top, \quad \text{var}[\boldsymbol{\mu}] \propto \frac{\mathbb{E}[\boldsymbol{\Sigma}]}{1/2\kappa}. \tag{18}$$

To generate these predictions, this method optimizes the analytical solution of the likelihood function:

$$\mathbb{P}(\boldsymbol{y}_i \mid \omega_i) = t_{\nu_i - N + 1}\left(\boldsymbol{y}_i \mid \boldsymbol{\mu}_{0i}, \frac{1}{\nu_i - N + 1} \frac{1 + \kappa_i}{\kappa_i} \nu_i \boldsymbol{L_i} \boldsymbol{L_i}^T\right). \tag{19}$$

However, it lacks mechanisms to ensure a structured and continuous latent space, which is important to leverage genetic structural similarity. Additionally, some studies [29, 30] have noted that the model takes the risk of halting the optimization process in highly uncertain areas, denoted as "zero-evidence".

# F Derivation and Proof

This section includes theoretical derivations and proofs of PRESCRIBE. First, § F.1 provides the derivation for the expected log-likelihood ($\mathcal{L}_1$) of a multivariate Gaussian distribution under a Normal-Inverse-Wishart prior. The derivation for the differential entropy of the Normal-Wishart distribution ($\mathcal{L}_2$) is presented in § F.2. § F.3 offers a justification for how Eq. 3.4 mitigates the issue of zero gradients in regions of high uncertainty. Furthermore, a multivariate approximation for the Gamma and Digamma functions is derived in § F.4. Moreover, this section also presents a proof demonstrating that the pseudo E-distance preserves the ranking order of the true E-distance in § F.5. Finally, we review the mechanism that NatPN uses to capture generalization difficulty. § F.6

## F.1 Derivation of $\mathcal{L}_1$

**Expected Log-Likelihood $\mathcal{L}_1$.** $\mathcal{L}_1$ optimizes expected log-likelihood under predicted posterior $\boldsymbol{\omega}$, improving accuracy and reducing uncertainty. It also affects other data points' evidence due to flow normalization. Its closed form is as follows:

$$
-\frac{N}{2}\ln(2\pi) + \frac{1}{2}\left(\ln|2\nu_i \boldsymbol{L}_i \boldsymbol{L}_i^T| + \psi_N\left(\frac{\nu_i}{2}\right)\right)
\tag{20}
$$
$$
-\frac{1}{2}\left(\left(\boldsymbol{y}_i - \boldsymbol{\mu}_{0x_i}\right)\left(\boldsymbol{y}_i - \boldsymbol{\mu}_{0x_i}\right)^T \nu_i^2 \boldsymbol{L}_i \boldsymbol{L}_i^T + \frac{\boldsymbol{I}}{2\nu_i}\right)
$$

For stability, $\psi_N(x)$ is approximated by $\sum_{n=1}^{N}\ln(\frac{x-n+1}{2})$ (Appx. F.4).

The Normal-Inverse-Wishart (NIW) distribution $\boldsymbol{\mu}, \boldsymbol{\Sigma} \sim \mathbb{W}^{-1}(\boldsymbol{\mu}_0, \kappa, \nu, \boldsymbol{\Psi})$ is the conjugate prior of the normal distribution $\boldsymbol{y} \sim \mathbb{N}(\boldsymbol{\mu}, \boldsymbol{\Sigma})$. Note that both parameters $\kappa$ and $\nu$ can be viewed as pseudo counts. However, PRESCRIBE enforces a single pseudo-count $n$ corresponding to $\kappa = 2\nu$.

**Target Distribution.** The density and the entropy of the Multivariate Normal distribution are:

$$
\mathbb{N}(\boldsymbol{\mu}, \boldsymbol{\Sigma}) = \frac{\exp\left(-\frac{1}{2}(\mathbf{x} - \boldsymbol{\mu})^{\mathrm{T}}\boldsymbol{\Sigma}^{-1}(\mathbf{x} - \boldsymbol{\mu})\right)}{\sqrt{(2\pi)^N|\boldsymbol{\Sigma}|}}
\tag{21}
$$

$$
\mathbb{H}[\mathbb{N}(\boldsymbol{\mu}, \boldsymbol{\Sigma})] = \frac{N}{2}\ln(2\pi e) + \frac{1}{2}\ln|\boldsymbol{\Sigma}|
\tag{22}
$$

**Conjugate Prior Distribution.** The density of the NIW distribution is:

$$
\mathbb{W}^{-1}(\boldsymbol{\mu}, \boldsymbol{\Sigma} \mid \boldsymbol{\mu}_0, \kappa, \nu, \boldsymbol{\Psi}) = \mathcal{N}\left(\boldsymbol{\mu} \mid \boldsymbol{\mu}_0, \frac{1}{\kappa}\boldsymbol{\Sigma}\right)\mathbb{W}^{-1}(\boldsymbol{\Sigma} \mid \boldsymbol{\Psi}, \nu),
\tag{23}
$$

$$
\mathbb{W}^{-1}(\boldsymbol{\Sigma} \mid \boldsymbol{\Psi}, \nu) = \frac{|\boldsymbol{\Psi}|^{\nu/2}|\boldsymbol{\Sigma}|^{-\frac{\nu+N+1}{2}}}{2^{\frac{\nu N}{2}}\Gamma_N\left(\frac{\nu}{2}\right)}\exp\left\{-\frac{1}{2}\operatorname{tr}\left(\boldsymbol{\Sigma}^{-1}\boldsymbol{\Psi}\right)\right\}.
\tag{24}
$$

The full version of the probability density function is:

$$
\mathbb{W}^{-1}(\boldsymbol{\mu}, \boldsymbol{\Sigma} \mid \boldsymbol{\mu}_0, \kappa, \nu, \boldsymbol{\Psi}) = \frac{\kappa^{N/2}|\boldsymbol{\Psi}|^{\nu/2}|\boldsymbol{\Sigma}|^{-\frac{\nu+N+2}{2}}}{(2\pi)^{N/2}2^{\frac{\nu N}{2}}\Gamma_N\left(\frac{\nu}{2}\right)}\exp\left\{-\frac{1}{2}\operatorname{Tr}\left(\boldsymbol{\Psi}\boldsymbol{\Sigma}^{-1}\right) - \frac{\kappa}{2}\left(\boldsymbol{\mu} - \boldsymbol{\mu}_0\right)^T\boldsymbol{\Sigma}^{-1}\left(\boldsymbol{\mu} - \boldsymbol{\mu}_0\right)\right\},
\tag{25}
$$

where $\boldsymbol{\Gamma}_N$ is the multivariate gamma function and $\operatorname{Tr}(\cdot)$ is the Trace of the given matrix.

**Expected Log-Likelihood.** The expected likelihood of the Multivariate Normal distribution $\mathbb{N}(\boldsymbol{\mu}, \boldsymbol{\Sigma})$ under the NIW distribution $\mathbb{W}^{-1}(\boldsymbol{\mu}_0, \kappa, \nu, \boldsymbol{\Psi})$ is:

$$\mathbb{E}_{(\boldsymbol{\mu}, \boldsymbol{\Sigma}) \sim \mathbb{W}^{-1}(\boldsymbol{\mu}_0, \kappa, \nu, \boldsymbol{\Psi})}[\ln \mathbb{N}(y \mid \boldsymbol{\mu}, \boldsymbol{\Sigma})] \tag{26}$$

$$= \mathbb{E}\left[-\frac{N}{2}\ln(2\pi) - \frac{1}{2}\ln|\boldsymbol{\Sigma}| - \frac{1}{2}(\mathbf{y}_i - \boldsymbol{\mu})^T \boldsymbol{\Sigma}^{-1}(\mathbf{y}_i - \boldsymbol{\mu})\right] \tag{27}$$

$$= \frac{1}{2}\left(-\mathbb{E}\left[(\mathbf{y}_i - \boldsymbol{\mu})^T \boldsymbol{\Sigma}^{-1}(\mathbf{y}_i - \boldsymbol{\mu})\right] - \mathbb{E}\left[\ln|\boldsymbol{\Sigma}|\right] - N\ln(2\pi)\right) \tag{28}$$

$$= \frac{1}{2}\left(-\mathbb{E}\left[\mathrm{Tr}\left[(\mathbf{y}_i - \boldsymbol{\mu})^T(\mathbf{y}_i - \boldsymbol{\mu})\boldsymbol{\Sigma}^{-1}\right]\right] + \mathbb{E}\left[\ln|\boldsymbol{\Sigma}^{-1}|\right] - N\ln(2\pi)\right) \tag{29}$$

$$= \frac{1}{2}\left(-\mathbf{y}_i^T \mathbf{y}_i \mathbb{E}\left[\boldsymbol{\Sigma}^{-1}\right] + \mathbf{y}_i^T \mathbb{E}\left[\boldsymbol{\mu}\boldsymbol{\Sigma}^{-1}\right] + \mathbf{y}_i \mathbb{E}\left[\boldsymbol{\mu}^T\boldsymbol{\Sigma}^{-1}\right] - \mathbb{E}\left[\boldsymbol{\mu}^T\boldsymbol{\mu}\boldsymbol{\Sigma}^{-1}\right]\right)$$
$$+ \frac{1}{2}\left(\left(N\ln 2 - \ln|\boldsymbol{\Psi}| + \psi_N\left(\frac{\nu}{2}\right)\right) - N\ln(2\pi)\right) \tag{30}$$

$$= -\frac{1}{2}\left((\boldsymbol{y}_i - \boldsymbol{\mu}_0)^T(\boldsymbol{y}_i - \boldsymbol{\mu}_0)\nu^2\boldsymbol{L}\boldsymbol{L}^T + \frac{1}{2\nu}\boldsymbol{I} + \ln|2\nu\boldsymbol{L}\boldsymbol{L}^T| + \psi_N\left(\frac{\nu}{2}\right) - N\ln(2\pi)\right). \tag{31}$$

Here $\psi_N(\cdot)$ denotes the multivariate digamma function. In PRESCRIBE's formulation, we can obtain $\boldsymbol{\Psi}^{-1} = \nu\boldsymbol{L}\boldsymbol{L}^T$ and the moment of the NIW distribution $\mathbb{E}\left[\boldsymbol{\mu}\boldsymbol{\Sigma}^{-1}\right] = \boldsymbol{\mu}_0 \cdot \nu\boldsymbol{L}\boldsymbol{L}^T$, $\mathbb{E}\left[\boldsymbol{\Sigma}^{-1}\right] = \nu\boldsymbol{L}\boldsymbol{L}^T$, $\mathbb{E}\left[\boldsymbol{\mu}^T\boldsymbol{\mu}\boldsymbol{\Sigma}^{-1}\right] = \boldsymbol{\mu}_0 \cdot \nu\boldsymbol{L}\boldsymbol{L}^T + \frac{1}{\kappa}\boldsymbol{I}$, and the moment of the Inverse Wishart distribution is

$$\mathbb{E}\left[\ln|\boldsymbol{\Sigma}|\right] = -N\ln 2 + \ln|\boldsymbol{\Psi}| - \psi_N\left(\frac{\nu}{2}\right) = \ln\left|\frac{\boldsymbol{\Psi}}{2}\right| - \psi_N\left(\frac{\nu}{2}\right). \tag{32}$$

### F.2 Derivation of $\mathcal{L}_2$

**Entropy Regularization $\mathcal{L}_2$.** $\mathcal{L}_2$ is an entropy prior favoring high-entropy posterior distributions $\mathbb{W}_i^{\text{post}}$. Its closed form:

$$-\frac{N+1}{2}\ln|2\nu_i\boldsymbol{L}_i\boldsymbol{L}_i^T| + \ln\Gamma_N\left(\frac{\nu_i}{2}\right)$$
$$-\frac{\nu_i + N + 1}{2}\psi_N\left(\frac{\nu_i}{2}\right) + \frac{\nu_i N}{2} \tag{33}$$

Approximations for stability (Appx. F.4): $x\psi(x) \approx x\ln(x) - \frac{1}{2}$ and $\ln\Gamma_N(x) \approx \frac{N(N-1)}{4}\ln(2\pi) + \frac{1}{2}\sum_{n=1}^N\left(\ln 2\pi - (x+1-n) + (x-n)\ln(\frac{x+1-n}{2})\right)$. In this section, F.1 gives the derivation of the expected log-likelihood of a multivariate Gaussian distribution under Normal Inverse Wishart prior Eq.20. F.2 presents the derivation of the differential entropy of the normal Wishart distribution Eq.33. In F.3, we justify that Eq.3.4 can avoid zero gradient in highly uncertain areas. Lastly, we derive an approximation of Gamma and Digamma in a multivariate version in F.4. Before deriving Eq.33, we first introduce two propositions as follows:

**Proposition 1.** *For $\boldsymbol{\Lambda} \sim \mathbb{W}(\boldsymbol{\Psi}, \nu)$ and any positive definite matrix $A \in \mathbb{R}^{N \times N}$,*

$$\mathbb{E}[\mathrm{tr}(\boldsymbol{\Lambda}A)] = \nu\,\mathrm{tr}(\boldsymbol{\Lambda}A). \tag{34}$$

*Proof.* $\mathbb{E}[\mathrm{tr}(\boldsymbol{\Lambda}A)] = \mathrm{tr}(\mathbb{E}[\boldsymbol{\Lambda}]A) = \nu\,\mathrm{tr}(\boldsymbol{\Psi}A)$. □

**Proposition 2.** *For $\boldsymbol{\Sigma} \sim \mathbb{W}^{-1}(\boldsymbol{\Psi}, \nu)$ and any positive definite matrix $A \in \mathbb{R}^{N \times N}$,*

$$\mathbb{E}[\mathrm{tr}(\boldsymbol{\Sigma}^{-1}A)] = \nu\,\mathrm{tr}(\boldsymbol{\Psi}^{-1}A). \tag{35}$$

*Proof.* By definition, $\boldsymbol{\Sigma}^{-1} \sim \mathbb{W}(\boldsymbol{\Psi}^{-1}, \nu)$, so according to Proposition 1, we can yield: $\mathbb{E}[\mathrm{tr}(\boldsymbol{\Sigma}^{-1}A)] = \nu\,\mathrm{tr}(\boldsymbol{\Psi}^{-1}A)$. □

**The Differential Entropy of the Inverse Wishart Distribution.** Using the Inverse Wishart density given in Eq.24, the Inverse Wishart differential entropy is:

$$\mathbb{H}(\mathbf{\Sigma}) = -\mathbb{E}[\ln \mathbb{W}^{-1}(\mathbf{\Sigma} \mid \mathbf{\Psi}, \nu)] \tag{36}$$

$$= -\frac{\nu}{2} \ln |\mathbf{\Psi}| + \frac{\mathbb{E}\left[\mathrm{tr}\left(\mathbf{\Sigma}^{-1} \mathbf{\Psi}\right)\right]}{2} + \frac{\nu N}{2} \ln 2 + \ln \Gamma_N \left(\frac{\nu}{2}\right) + \frac{\nu + N + 1}{2} \mathbb{E}[\ln |\mathbf{\Psi}|] \tag{37}$$

$$\overset{(a)}{=} -\frac{\nu}{2} \ln |\mathbf{\Psi}| + \frac{\nu \,\mathrm{tr}\left(\mathbf{\Psi}^{-1} \mathbf{\Psi}\right)}{2} + \frac{\nu N}{2} \ln 2 + \ln \Gamma_N \left(\frac{\nu}{2}\right) + \frac{\nu + N + 1}{2} \left(\ln |\mathbf{\Psi}| - N \ln 2 - \psi_N \left(\frac{\nu}{2}\right)\right) \tag{38}$$

$$\overset{(b)}{=} \frac{N+1}{2} \ln \left|\frac{\mathbf{\Psi}}{2}\right| + \frac{\nu N}{2} + \ln \Gamma_N \left(\frac{\nu}{2}\right) - \frac{\nu + N + 1}{2} \psi_N \left(\frac{\nu}{2}\right), \tag{39}$$

where $(a)$ uses Proposition 2 and Eq.32, and in $(b)$ used $\mathrm{tr}(\mathbf{\Psi}^{-1} \mathbf{\Psi}) = \mathrm{tr}(\boldsymbol{I}) = N$ and simplified.

### F.3 Derivation of $\mathcal{L}_4$

**Zero Gradient Regions.** Research has shown that when the evidence approaches zero, the gradient will also become zero, resulting in stopping optimisation.

**Proposition 3.** *The model cannot learn from samples in high uncertainty areas by optimizing $\mathcal{L}_1$.*

*Proof.* Given the parameters $p_\nu$ of the flow module, we can obtain $\partial \nu^{post}$ as Eq.8. Therefore, the gradient of the loss function $\mathcal{L}_1$ with respect to $p_\nu$ is given by:

$$\frac{\partial \mathcal{L}_1}{\partial p_\nu} = \frac{\partial \mathcal{L}_1}{\partial \nu^{post}} \frac{\partial \nu^{post}}{\partial p_\nu}$$

$$= \frac{\partial \mathcal{L}_1}{\partial \nu^{post}} \cdot N \cdot \frac{e^{p_\nu + N_H}}{(e^{p_\nu + N_H} + \nu^{prior})^2} \tag{40}$$

$$= \frac{\partial \mathcal{L}_1}{\partial \nu^{post}} \cdot N \cdot \underbrace{\frac{1}{e^{p_\nu + N_H} + 2\nu^{prior} + \nu^{prior} e^{-(p_\nu + N_H)}}}_{(a)}. \tag{41}$$

In a high uncertainty areas, $p_\nu \to -\infty \Rightarrow (a) \to 0$, so it leads to zero gradient with respect to $p_\nu$. The model cannot learn from samples in these regions. $\square$

Then, we prove that the effectiveness of the proposed $\mathcal{L}_4$ can be supervised not only by pairwise distance but also by avoidance of the zero gradient scenario.

**Proposition 4.** *$\mathcal{L}_4$ can help the model learn from samples within zero-gradient regions.*

*Proof.* Through Eq.8, we can obtain $\nu^{post} = \ln(\frac{p_\nu - N}{2N - p_\nu} \cdot \nu^{prior})$. Then we can derive the gradient of $\mathcal{L}_4$ with respect to $p_\nu$ is given by:

$$-\frac{\partial \mathcal{L}_4}{\partial p_\nu} = -\frac{\partial \mathcal{L}_4}{\partial \nu^{post}} \frac{\partial \nu}{\partial p_\nu}$$

$$= -\left|\boldsymbol{y}_i - \boldsymbol{\mu}_{0x_i}\right| \frac{N}{(p_\nu - N)(2N - p_\nu)} \cdot N \cdot \frac{(p_\nu - N)(2N - p_\nu)}{N^2} \tag{42}$$

$$= -\left|\boldsymbol{y}_i - \boldsymbol{\mu}_{0x_i}\right| \neq 0. \tag{43}$$

$\square$

The gradient remains non-zero when $p_\nu \to -\infty$, which avoids zero gradients in these areas.

## F.4 Approximation of Multivariate Gamma and Digamma Function.

The computation of a distribution's entropy often requires subtracting large numbers from each other. Although these numbers tend to be very close together, this introduces numerical challenges. For large parameter values, we approximate the entropy by substituting numerically unstable terms and simplifying the resulting formula. For this procedure, we use the following equivalences [31]:

$$\ln \Gamma(x) \approx \frac{1}{2} \ln 2\pi - x + \left( x - \frac{1}{2} \right) \ln x, \tag{44}$$

$$\psi(x) = \ln x - \frac{1}{2x} + \mathcal{O}\left( \frac{1}{x^2} \right) \approx \ln x. \tag{45}$$

According to the definitions of the multivariate Gamma functions and the multivariate Digamma Function:

$$\ln \Gamma_N(x) = \frac{N(N-1)}{4} \ln(2\pi) + \sum_{n=1}^{N} \ln \left( \Gamma \left( \frac{x+1-n}{2} \right) \right), \tag{46}$$

$$\psi_N(x) = \sum_{n=1}^{N} \psi \left( \frac{x-n+1}{2} \right), \tag{47}$$

we can obtain multivariate approximation versions as follows:

$$\ln \Gamma_N(x) \approx \frac{N(N-1)}{4} \ln(2\pi) + \frac{1}{2} \sum_{n=1}^{N} \left( \ln 2\pi - (x+1-n) + (x-n) \ln(\frac{x+1-n}{2}) \right), \tag{48}$$

$$\psi_N(x) \approx \sum_{n=1}^{N} \ln(\frac{x-n+1}{2}). \tag{49}$$

## F.5 Proof of Ranking Order Preservation

**Theorem 1.** *Under fixed prior conditions, $\hat{E}$ preserves the ranking order of $E$.*

*Proof.* Under min-max normalization with fixed control states, we derive: $E = 2\delta_{XY} - Y$, where $\delta_{XY}, Y \in [N, 2N]$. Through model design constraints: distance from prior $\nu \propto \delta_{XY}$ and target entropy $\mathbb{H} \propto Y$. Let $\nu = a\delta_{XY}$ and $\mathbb{H} = bY$ with $a, b > 0$. Then:

$$\hat{E} = 2 \frac{a(\delta_{XY} - N)}{aN} - \frac{b(Y - N)}{bN}$$
$$= 2 \frac{\delta_{XY} - N}{N} - \frac{Y - N}{N}$$
$$= \frac{2\delta_{XY} - Y - N}{N} = \frac{1}{N} E - 1$$

Let $C = -1$. Then $\hat{E} = \frac{1}{N} E + C$. Since $N$ is a positive constant (as $\delta_{XY}, Y \in [N, 2N]$ implies $N > 0$), $\frac{1}{N}$ is a positive constant. This linear relationship, $\hat{E} = kE + C$ where $k = \frac{1}{N} > 0$, preserves ordinal rankings between $\hat{E}$ and $E$. If $E_1 < E_2$, then $kE_1 < kE_2$, and $kE_1 + C < kE_2 + C$, which implies $\hat{E}_1 < \hat{E}_2$. Thus, the ranking order is preserved. □

## F.6 NatPN

The authors of NatPN [9] have proved that PRESCRIBE can estimate evidence that asymptotically approaches zero under OOD conditions (e.g., far from the training set) as follows:

**Lemma 1.** *Let $\{Q_l\}_l^R$ be the set of linear regions associated to the piecewise ReLU network $f_\phi(x)$. For any $x \in \mathbb{R}^D$, there exists $\delta^* \in \mathbb{R}^+$ and $l^* \in \{1, \ldots, R\}$ such that $\delta x \in Q_{l^*}$ for all $\delta > \delta^*$ [32].*

**Theorem 2.** *Let a NatPN model parametrized with a (deep) encoder $f_\phi$ with piecewise ReLU activations, a decoder $g_\psi$ and the density $\mathbb{P}(z|\omega)$. Let $f_\phi(x) = V^{(l)}x + a^{(l)}$ be the piecewise affine*

*representation of the ReLU network $f_\phi$ on the finite number of affine regions $Q^{(l)}$ [32]. Suppose that $V^{(l)}$ have independent rows and the density function $\mathbb{P}(z|\omega)$ has bounded derivatives, then for almost any $x$ we have $\mathbb{P}(f_\phi(\delta \cdot x)|\omega) \underset{\delta \to \infty}{\to} 0$. i.e the evidence becomes small far from training data.*

*Proof.* Let $x \in \mathbb{R}^D$ be a non-zero input and $f_\phi$ be a ReLU network. Lem. 1 implies that there exists $\delta^* \in \mathbb{R}^+$ and $l \in \{1, \ldots, R\}$ such that $\delta \cdot x \in Q^{(l)}$ for all $\delta > \delta^*$. Thus, $z_\delta = f_\phi(\delta \cdot x) = \delta \cdot (V^{(l)}x) + a^{(l)}$ for all $\delta > \delta^*$. Note that for $\delta \in [\delta^*, +\infty]$, $z_\delta$ follows an affine half line $S_x = \{z | z = \delta \cdot (V^{(l)}x) + a^{(l)}, \delta > \delta^*\}$ in the latent space. Further, note that $V^{(l)}x \neq 0$ and $\|z_\delta\| \underset{\delta \to \infty}{\to} +\infty$ since $x \neq 0$ and $V^{(l)}$ has independent rows. $\qquad\square$

Our model adaptation did not conflict with its architectural prerequisite.

## G  Dataset Details

Our datasets are downloaded from scPerturb [8] and perform the following preprocessing steps:

**Quality control.** To ensure consistent and homogeneous quality throughout the different data sets, we filter cells with fewer than $1,000$ and genes with fewer than $50$ cells per data set.

**Feature Selection.** We identified the top $2,000$ highly variable genes (HVGs) from each dataset using standard procedures and included all perturbed genes in the dataset's feature list.

**Normalization.** The raw count data were normalized using the lognormalization method, implemented through the standard preprocessing workflow of Scanpy [33].

**Identifying Differentially Expressed Genes (DEGs).** We identified DEGs by comparing perturbed cells to control cells for each perturbation. Genes were ranked based on adjusted p-values computed using the Wilcoxon test in Scanpy, with those p-values$< 0.05$ designated as DEGs.

**PCA.** Considering that too large a dimension will cause the Student's t-distribution to not be distinguishable from the Gaussian distribution, we choose the PCA components as $10$. The PCA was trained on the training set and performed on the validation set and test set. Additionally, we save the PCA model for reconstructing all gene' values. The results are saved in adata.obsm[$'y\_pca'$].

**E-distance Calculation.** The E-distance for each perturbation was calculated to quantify the effect of the perturbation. E distance is a statistical measure of distance between two distributions, which is utilized to indicate the degree of perturbation effect in scPerturb. We utilized the scPerturb [8] Python library to compute this metric between the control and perturbed cells. For each perturbation, we subsampled all perturbations to their greatest common divisors to perform the calculation. The distance e, pairwise distance, and self-distance are saved in adata.obsm[$"n''"$], adata.obsm[$'d'$] and adata.obsm[$'s'$], respectively. For normalization, we perform the max-min normalisation on all the training sets' E distances onto $[N, 2N]$.

**Spilt Dataset.** The filtered perturbations were divided into training, testing, and validation sets according to the default operation in GEARS [1]. The training set was used to fit the models, the validation set was used to select the best model, and the testing set was used to evaluate the final model performance. And for more details, see Tab.7.

## H  Baseline Setting Details

To evaluate PRESCRIBE's effectiveness, we benchmarked it against seven baseline models and five PRESCRIBE variants. Unless otherwise specified, all baseline model parameters were configured according to this benchmark [6] research to ensure a fair comparison.

Table 7: Summary Statistics of Different Datasets

| Metric | Norman2019 [4] | Replogle2022_Rep1 [5] | Replogle2022_K562 [5] |
|---|---|---|---|
| Average E-distance | 59.172741 | 135.737438 | 31.004867 |
| Number train/val/test | 80712/4758/6009 | 121242/14959/37863 | 133412/13593/45114 |
| Type of train/val/test | 213/32/32 | 1036/115/384 | 734/82/272 |
| Test Description | Only double gene knock-out perturbation | Only single gene knock-out perturbation | Only single gene knock-out perturbation |
| UMI Count | 2037 | 3212 | 2868 |
| Cell Type | K562 | RPE1 | K562 |

## H.1 AverageKnown

AverageKnown is a fundamental approach that averages gene expression across all cells within known perturbations to predict unseen perturbations.

## H.2 Linear

The linear method is derived from a benchmark study [20]. The optimization objective is given by

$$\arg\min_{\mathbf{W}} \left\| \mathbf{Y}^{\text{train}} - \left( \mathbf{G}\mathbf{W}\mathbf{P}^T + \boldsymbol{b} \right) \right\|_2^2, \tag{50}$$

where $\mathbf{Y}^{\text{train}}$ is the gene expression matrix of perturbed cells. The rows represent the gene feature set, and the columns represent the perturbed genes in the training data. $G$ is the gene embedding matrix obtained from the top $K$ principal components derived from principal component analysis (PCA) on $\mathbf{Y}^{\text{train}}$. $P$ is a subset of $G$ that includes the perturbed genes in the training data. The fitted matrix is expressed as $\boldsymbol{b} = \frac{1}{N} \sum_{i=1}^{N} \mathbf{Y}_{:i}^{\text{train}}$, where $b$ is the vector of average gene expressions from $\mathbf{Y}^{\text{train}}$. The matrix $W$ is solved using the normal equations:

$$\mathbf{W} = \left( \mathbf{G}^T \mathbf{G} + \lambda \mathbf{I} \right)^{-1} \mathbf{G}^T \left( \mathbf{Y}^{\text{train}} - \boldsymbol{b} \right) \mathbf{P} \left( \mathbf{P}^T \mathbf{P} + \lambda \mathbf{I} \right)^{-1}. \tag{51}$$

Here, we set $k$ to $512$ and $\lambda$ to $0.1$. The fitted $W$ is then used for prediction:

$$\widehat{\mathbf{Y}} = \boldsymbol{b} + \mathbf{G}\mathbf{W}\mathbf{P}^T, \tag{52}$$

where $\mathbf{Y}$ represents the predicted gene expression, and $P$ denotes the subset of $G$ that corresponds to the embeddings of perturbed genes in the testing data.

## H.3 Linear-GPT

Linear-scGPT [20] employs the same modeling approach as the linear method, with the key distinction that the gene embedding matrix $G$ is derived from the scGPT model rather than the PCA components of the training data. We followed the instructions from the GitHub repository of the linear method to obtain the scGPT gene embeddings.

## H.4 scGPT

The scGPT [10] model is versatile across various scenarios due to its ability to accept any length of gene input. In the unseen perturbation transfer and unseen cell type transfer scenarios, we adhered to the guidance in the scGPT GitHub repository to reformat data and train models. Default parameter settings were employed, with the model trained using all genes in the feature set. Similar to GEARS, we trained the scGPT model for 20 epochs and selected the best-performing model based on validation results for testing. For zero-shot transfer and cell state transition scenarios, the scGPT model was trained on a combined dataset comprising a total of 17 datasets, ensuring broader generalization across different biological contexts.

## H.5 scFoundation

scFoundation [15] also employs transformer-based architectures, making it suitable for in silico perturbation tasks, similar to the training paradigm of scGPT. Specifically, we leveraged the encoder module of the scFoundation model to train the perturbation model. The training and testing procedures mirrored those in scGPT. Since scFoundation incorporates more transformer layers than scGPT, we set the learning rate to $1 \times 10^{-5}$.

### H.6 CellOracle

We used the code provided on the CellOracle [12] GitHub repository. Since our control group data contained significantly fewer cells compared to the tutorial dataset, we reduced 'n_iters' from $100,000$ to $1,000$ to improve computational efficiency.

### H.7 GEARS

**Standard.** We followed the tutorials provided in the GEARS [1] GitHub repository. GEARS is designed for scenarios involving unseen perturbations. We first reformatted our data to align with GEARS' input requirements. The model was then trained using its default parameter settings, with the number of epochs set to 20. The best-performing model, selected based on its performance on the validation dataset, was used to evaluate the test dataset.

**GEARS-Drop.** GEARS provides an additional version that predicts the variance of each gene under specified perturbations, utilizing a loss function based on negative log likelihood as follows:

$$L_{\text{unc}} = \frac{1}{T} \sum_{k=1}^{T} \frac{1}{T_k} \sum_{l=1}^{T_k} \frac{1}{G} \sum_{u=1}^{K} \exp\left(-s_u\right) \left(\mathbf{g}_u - \hat{\mathbf{g}}_u\right)^{(2+\gamma)} \tag{53}$$

where $\hat{\mathbf{g}}_u$ is the predicted post-perturbation scalar and $\hat{\sigma}_u^2$ is the variance, $s_u = \log \hat{\sigma}_u^2 = \mathbf{w}_u^{\text{unc}} \mathbf{h}_u^{\text{post-pert}} + b_u^{\text{unc}}$.

**GEARS-Ens.** We ensemble GEARS-Drop by employing different training/validation splits and evaluating on the same test set. The results are averaged for both the predicted values and the logarithmic variance. We set the ensemble number to 5.

### H.8 Variants Of PRESCRIBE

**Ours-NOINFO.** We did not use the control status's mean and covariance to initialize the prior distribution. Instead, we adopt zeros as the prior mean and a zero matrix as the prior covariance.

**Ours-Null.** Ours-Null represents our model's untrained state.

**Ours-Drop.** The Ours-Drop variant utilizes the same encoder architecture as PRESCRIBE but replaces the decoder with Multi-Layer Perceptrons (MLPs). It employs the same loss function formulation as GEARS-Drop.

**Ours-Ens.** We ensemble Ours-Drop by employing different training/validation splits and evaluating on the same test set. The results are averaged for both the predicted values and the logarithmic variance. We set the ensemble number to 5.

**Ours-MLPs.** The Ours-MLPs variant is based on Ours-Drop but incorporates an additional mean squared error loss term. It uses two linear layers with a ReLU activation function to regress the E-distance from the latent embeddings.

## I Expected Calibration Error Results

We evaluate the uncertainty calibration of our model using the Expected Calibration Error (ECE). As shown in Tab. 8, PRESCRIBE consistently achieves lower ECE scores across all three datasets compared to the baselines, indicating better-calibrated uncertainty estimates.

## J Limitations

**Input Embeddings.** The model's performance relies on the quality of input embeddings from foundation models. Applying it to complex chemical perturbations, which are more intricate than genetic ones, requires developing correspondingly richer embeddings.

| Model | Norman | Rep1 | K562 |
|---|---|---|---|
| **PRESCRIBE (Ours)** | **30.59** | **50.38** | **41.47** |
| GEARS-unc | 32.45 | 63.96 | 64.76 |
| GEARS-Ens | 38.63 | 53.85 | 43.07 |

Table 8: Expected Calibration Error (ECE), multiplied by 100. Lower values indicate better calibration. Our model, PRESCRIBE, is compared against two baselines.

**Continuous Data Assumption.** We assume a continuous Gaussian space, while single-cell data is often represented as discrete counts. A potential solution is to use an autoencoder (e.g., scVI [34]) to project count data into a continuous latent space, apply our method, and then reconstruct the counts.

**Representation of Epistemic Uncertainty.** Our framework approximates epistemic uncertainty with latent space density. While this is a practical approach, it is an indirect measurement, as genuine epistemic uncertainty is difficult to quantify and interpret [35]. Developing more direct and robust methods for representing epistemic uncertainty is a crucial direction for future research.

## K   Broader Impacts

Our framework has broader applicability beyond its current focus on predicting gene perturbations. Many disciplines within the AI for Science landscape, such as materials science or drug discovery, face two common challenges: models are usually required to make predictions on novel, out-of-distribution inputs, and the training data is often derived from experimental observations subject to significant inherent uncertainty. By providing a unified approach to quantifying both epistemic (out-of-distribution) and aleatoric (data) uncertainty, our method can enhance the reliability and trustworthiness of machine learning models in these critical scientific applications.

