# OpenReview forum: "PRESCRIBE: Predicting Single-Cell Responses with Bayesian Estimation"
_NeurIPS.cc/2025/Conference — NeurIPS 2025 poster_

### Official Review · Reviewer_Z96M · 2025-06-28

**Clarity:** 2
**Significance:** 2
**Originality:** 1
**Rating:** 3
**Confidence:** 4

**Summary:**

The paper proposes a framework called PRESCRIBE for predicting responses to unseen gene perturbations in single-cell RNA sequencing (scRNA-seq) data using Bayesian estimation. The PRESCRIBE framework employs the previously proposed Natural Posterior Network (NatPN) to provide uncertainty estimates via a normal-inverse-Wishart distribution and incorporates scGPT for gene embeddings. Additionally, the paper introduces a pseudo E-distance as a confidence score derived from the multivariate NatPN, which can be computed for unseen perturbations. Experimental results on three-gene knockout datasets demonstrate that i) the pseudo E-distance score is correlated with the empirical E-distance, and ii) PRESCRIBE achieves performance competitive with baseline methods in terms of correlation-based evaluation metrics for inferring unseen gene perturbation transcriptional profiles.

**Questions:**

- What is the impact of using scGPT for gene embeddings compared to alternative gene embedding approaches, e.g., ESM-2, and GNN specified according to the GEARS formulation?
- What's the impact of using the normal-inverse-Wishart distribution versus the Gamma-Poisson distribution for the Negative Binomial used in scVI?
- Why is the sum aggregation function is used for combined perturbations (Eq 7) instead of alternative set aggregation approaches, e.g., set transformers?
- Could you provide comprehensive non-correlation metrics, such as RMSE?

**Ethical Concerns:**

["NO or VERY MINOR ethics concerns only"]

**Limitations:**

- The paper discusses some limitations regarding the use of scGPT for gene embeddings; however, it should also address the potential biases introduced by the preprocessing approaches, such as high variable gene selection and log-normalizing the counts data.

**Quality:**

1

**Strengths And Weaknesses:**

**Strengths**

- The PRESCRIBE framework enables uncertainty estimation of model predictions.
- The paper provides source code, which enhances reproducibility.

**Weaknesses**

- *Originality/Significance* - The proposed approach appears to be a straightforward combination of previously proposed methods, namely NatPN, scGPT, and E-distances, with minimal modifications and without justification compared to potential alternative approaches:
1) What is the impact of using scGPT for gene embeddings compared to alternative gene embedding approaches, e.g., ESM-2, and GNN specified according to the GEARS formulation?
2) What's the impact of using the normal-inverse-Wishart distribution versus the Gamma-Poisson distribution for the Negative Binomial used in scVI?

- *Quality/Clarity* - The model formulation (section 3.3) should be expanded to provide complete modeling details:
1) The use of scGPT for the gene perturbation embeddings should be highlighted; only two lines are provided to describe such an important modeling decision.
2) It's unclear why the sum aggregation function is used for combined perturbations (Eq 7) instead of alternative set aggregation approaches, e.g., set transformers.
3) It's unclear how the control profiles are encoded via $f(c)$ in Eqn 7. Also, how is $c$ selected for each $y$ given that only one is provided for a specific cell?
4) The paper should provide complete parameterization of the decoder and the normalizing flow functions.

- *Weak experiments* -  Given that the technical contributions are limited, the experimental setup does not seem to support the claims:
1) While the paper provides an ablation study for some of the loss components in Eqn 9 (Table 4), they don’t seem comprehensive across all datasets, and the default NatPN objective (L1 and L2) should be provided.
2) The paper reports correlation metrics as percentages, which is unconventional. These should be given in the standard range [-1, 1], ideally with tests of statistical significance.
3) Figure 4: While PRESCRIBE seems to have better calibration than GEAR, it does not appear to be well-calibrated. Additionally, the calibration data points are few, with PRESCRIBE resulting in larger error bars. The paper should also provide alternative metrics for measuring uncertainty, including the coefficient of variation.
4) Table 1: The paper claims that pseudo E-distance is correlated with E-distance; however, the correlation coefficient seems too small for it to be an effective measure of E-distance.
5) It's unclear why non-correlation-based metrics, such as RMSE, are not considered.

---

> ### Author Rebuttal · Authors · 2025-07-31
>
> Dear Reviewer Z96M,
>
> Thank you for your constructive feedback. We have addressed each of your points below.
>
> **Metrics and Their Descriptions**
> | Metric | Usage|Description|
> |------|------|------|
> | PCC1 | Performance Evaluation| The Pearson correlation between the predicted and true log fold change under specific perturbations.|
> | ACC1 | Performance Evaluation| The directional accuracy between the predictive and true log fold change under specific perturbations.|
> | RMSE | Performance Evaluation| Root mean square error between the predictive and true log fold change under specific perturbations.|
> | ACC2 | Caliberation Evaluation|The quintile classification accuracy between certain metrics and PCC1|
> | SPCC| Caliberation Evaluation| The Spearman correlation between certain metric and PCC1|
>
> > **Q1/W1.1/W2.1. Gene Embedding Ablation Study**
>
> Thank you for the suggestion. To clarify, the ESM model operates on amino acid sequences to produce protein embeddings, not gene embeddings, so a direct comparison is not applicable.
>
> However, to address the core of your question about the impact of the gene encoder, we conducted an ablation study comparing our model with three alternatives:
> 1. GEARS-GNN embeddings.
> 2. scFoundation embeddings.
>
> | Model | PCC1 | PCC1_DEG | ACC1 | ACC1_DEG |SPCC | ACC2 |
> | :--- | :--- | :--- | :--- | :--- |:---: |:---:  |
> | Ours (scFoundation) | 57.79 | 56.86 | 63.35 | 70.34 |37.02 | 22.58|
> | Ours (scFoundation)-5% | 59.68 | 58.36 | 63.75 | 70.56 |:---: |:--:|
> | Ours (scFoundation)-10% | 61.46 | 58.63 | 64.47 | 70.65 |:---: |:---:  |
> | Ours (GNN) | 14.31 | 44.47 | 50.90 | 66.94 |15.15 | 16.13 |
> | Ours (GNN)-5% | 14.58 | 45.29 | 50.87 | 66.90 |:---: |:---:  |
> | Ours (GNN)-10% | 14.76 | 45.62 | 51.03 | 67.04 |:---: |:---:|
> | Ours | 58.38 | 64.44 | 63.24 | 74.68 |67.19|38.71|
> | Ours-5% | 61.58 | 66.36 | 64.08 | 75.69 |:---: |:---:|
> | Ours-10% | 64.32 | 68.61 | 64.73 | 75.93 |:---: |:---:|
>
> > **Q2/W1.2: Choice of Distribution**
>
> We chose the Normal-Inverse-Wishart (NIW) distribution because our model, like the GEARS baseline, operates on preprocessed, normalized expression data, which is continuous. The NIW is the conjugate prior for a multivariate Gaussian likelihood, making it a natural fit for this data type.
>
> In contrast, the Negative Binomial distribution is designed for modeling discrete count data. While adapting our evidence-based framework to a discrete likelihood is a good proposal for future research, it would require a non-trivial theoretical derivation.
>
> > **Q3/W2.2/W2.3. Encoder Ablation Study**
>
> 1.  **Model Design:** Our encoder in Equation. 7 models, both an additive effect (first term) and a non-linear interaction effect (second term). This architecture was adopted from prior work [2] and chosen for its effectiveness and simplicity. The encoder module is flexible, requiring only a deterministic mapping from inputs (c, x) to latent embeddings z (Appx.F. 6, Theorem 2).
>
> 2.  **Experiment with a Set Transformer:** Following your suggestion, we implemented a version of our model using a Set Transformer as the encoder. The results are presented below.
>
> | Model | PCC1 | PCC1_DEG | ACC1 | ACC1_DEG | SPCC | ACC2 |
> | :--- | :---: | :---: | :---: | :---: | :---: | :---: |
> | Ours(Set) | 57.16 | 56.15 | 63.00 | 68.33 | 12.50 | 16.13 |
> | Ours(Set)-5% | 57.40 | 56.61 | 63.31 | 68.55 | :---: |:---:  |
> | Ours(Set)-10% | 57.67 | 57.79 | 63.35 | 68.62 | :---: |:---:  |
>
> Our detailed adaptation is
>
> $\sum f_{a1}(x_i) + f_{a2}(\sum f_{a1}(x_i)) \rightarrow f_{set}(CONCAT[f_{a1}(x_1),f_{a1}(x_2),...,f_{a12}(c)])$.
>
> While this is a promising direction, the Set Transformer did not outperform our original model on this dataset, potentially due to the large data requirements for training a representative transformer.
>
> > **Q4/W3.5: Comprehensive Non-Correlation Metrics (RMSE)**
>
> As requested, we now provide Root Mean Squared Error (RMSE) metrics (multiplied by 100).
>
> | Model | Norman-RMSE | Rep1-RMSE |K562-RMSE |
> | :--- | :--- | :--- | :--- |
> | GEARS | 10.52 | 17.85 | 15.42 |
> | Ours | 10.16 | 17.27 | 11.63 |
> | Ours-5% | 10.05 | 17.14 | 11.45 |
> | Ours-10% | 9.99 | 16.89 |11.40 |
>
> > **W1: Originality and Significance**
>
> The three major contributions of our work are
> 1. Motivation Originality. Our paper tries to solve overconfidence in single-cell response prediction tasks, a crucial yet often overlooked limitation in previous research.
> 2. Theoretical Derivation Originality. The derivation and application of a multivariate evidence-based framework for single-cell perturbation prediction. This includes the novel multivariate loss function and the associated parameter update mechanisms.
> 3. Model Originality. The newly designed model is expected to achieve uncertainty-aware single-cell responses prediction and is expected to benefit by increasing the accuracy of predictive models.
>
> > **W2.3: Control State Encoding**
>
> We apologize for the lack of clarity. The control profile $c_{i,j}$ for a given perturbation is the mean expression profile of all control cells in that experiment. We will state this explicitly in the revised manuscript.
>
> > **W2.4: Decoder and Normalizing Flow Parameterization**
>
> The overall architecture is described in Lines 150-158, and hyperparameters are listed in Appendix B.
>
> > **W3.1: Ablation Study on losses and other datasets**
>
> The $L_1$ term serves as the main regression target (like Mean Absolute Error for GEARS), so we did not ablate it.
>
> The effect of the $L_2$ regularization term is studied in the hyperparameter search (Figure 5b), where setting $\lambda_1 = 0$ corresponds to its ablation.
>
> The other ablations on the other two datasets are shown below:
>
> | Rpe1 | PCC | ACC | SPCC |
> | :--- | :--- | :--- | :--- |
> | Ours | 59.18 | 67.36 | 28.69 |
> | w/o L2 | 58.86 | 63.72 | -1.49 |
> | w/o L3 | 8.50 | 51.83 | 5.75 |
> | w/o L4 | 8.50 | 51.83 | 5.75 |
> | NO-INFO | 58.89 | 68.72 | -5.00 |
>
> | k562 | PCC | ACC | SPCC |
> | :--- | :--- | :--- | :--- |
> | Ours | 36.20 | 60.27 | 12.43 |
> | w/o L2 | 35.57 | 60.74 | -11.09 |
> | w/o L3 | 10.96 | 52.38 | -5.13 |
> | w/o L4 | 36.14 | 60.56 | 10.00 |
> | NO-INFO | 36.00 | 61.34 | -1.00 |
>
>
> > **W3.2: The paper reports correlation metrics as percentages, which is unconventional.**
>
> Regarding the calibration plot (Figure 5), the y-axis and x-axis show the percentile rather than PCC values, so it is not in the [-1, 1] interval.
>
> > **W3.3: While PRESCRIBE seems to have better calibration than GEAR, it does not appear to be well-calibrated. Additionally, the calibration data points are few, with PRESCRIBE resulting in larger error bars.**
>
> (1) We think that one limiting factor is the calculation of the ground-truth E-distance itself, which requires a large number of cells to be genuinely accurate, a number often unavailable in real datasets. However, the consistent trend of performance improvement upon filtering (as shown in Table 2) demonstrates that our uncertainty metric has already been put into practical use.
>
> (2) Actually, a large vertical error bar is a sign of good calibration. To conveniently visualize the data, we discretize continuous confidence scores and PCC values by binning them into percentile intervals, but variance within intervals is expected. Therefore, a better visualization might be to show 45-degree slanted error bars, which we are considering adopting.
>
> We updated more standard calibration metrics using Expected Calibration Error (ECE), a standard metric for calibrated regression (Kuleshov et al., ICML 2018). Lower ECE values indicate better calibration.
>
> The table below reports the ECE (multiplied by 100) for PRESCRIBE and baselines.
>
> | Model | Norman | Rep1 | K562 |
> | :--- | :---: | :---: | :---: |
> | Ours | 30.59 | 50.38 | 41.47 |
> | GEARS-unc | 32.45 | 63.96 | 64.76 |
> | GEARS-Ens | 38.63 | 53.85 | 43.07 |
>
> We will add the formal definition of ECE and a detailed discussion of these results to the revised manuscript.
>
> > **W3.4: Pseudo E-distance Correlation***
>
> Table 1(b) shows that the correlation between our predicted E-distance and the ground truth improves as the ground truth becomes more reliable (i.e., as $N$ increases). This suggests that the performance reported in Table 1(a) may be underestimated, as the number of available cells in the dataset limits the ground-truth E-distance calculation itself.

---

> > ### Author Response · Authors · 2025-08-05
> > **Look forward to your feedback**
> >
> > Dear Reviewer Z96M,
> >
> > Thank you for your time and for providing a detailed review of our manuscript. **Your comments regarding the ablation study for encoder model selection were very helpful. They prompted us to make this analysis more comprehensive. And remind us to clearly emphasize the model-agnostic feature of the encoder in our framework, which has strengthened the paper.**
> >
> > We have carefully addressed each of your points, supporting our responses with additional experiments. We hope our responses can resolve your concerns and would appreciate your feedback on our responses.
> >
> > Best regards,
> >
> > The Authors

---

> ### Comment · Reviewer_Z96M · 2025-08-06
> **Response by Reviewer Z96M**
>
> Thank you for the rebuttal and for providing additional results. The paper addresses some of my concerns but not all. I agree with reviewer *UDmR* that the clarity needs improvement, as the paper should be accessible to both ML and Biology audiences. Here are some of the outstanding issues:
>
> - The rebuttal claims that lines 150-158 and the Appendix specify both the Flow $f_{\psi} $ and Decoder functions $f_{\beta} $. Unfortunately, only text descriptions of what those functions are provided, not actual specifications. Could you provide the actual specifications?
> - The ablation study on the gene embeddings encoder is a good start. Note that ESM embeddings could also be used to extract gene embeddings since genes make proteins (see [1, 2]).
> - Since the focus of the paper is on uncertainty prediction, it's still unclear why the proposed Normal Inverse Wishart approach is better than comparable alternatives, such as CellFlow [2], which models discrete count data instead of log-normalized counts. Besides the ECE metric, could you also provide uncertainty estimation beyond correlation metrics?
> - Figure 4 and ECE metrics: Unfortunately, the proposed approach does not appear to be better calibrated than GEARS, although the ECE metric suggests some marginal improvement, which is unclear in terms of statistical significance.
> - Table 1: The paper claims that pseudo-E-distance is correlated with E-distance; however, the correlation coefficient seems too small for it to be an effective measure of E-distance. Could you clarify?
> - Also, in general, reporting correlation or ECE metrics as percentages is potentially misleading.
>
> Given the above outstanding issues, I am leaning more towards maintaining my score.
>
> *References*:
>
> - [1] Rosen et al. 2023, "Universal Cell Embeddings: A Foundation Model for Cell Biology"
> - [2] Klein et al. 2025, "CellFlow enables generative single-cell phenotype modeling with flow matching"

---

> ### Author Response · Authors · 2025-08-07
> **Issues with Direct Solutions**
>
> Dear Reviewer Z96M,
>
> Thank you for explicitly listing the unresolved issues. We appreciate the opportunity to clarify these points.
>
> We have structured our response into two categories based on your comments: **Issues with Direct Solutions** and **Issues Requiring Deeper Discussion**.
>
> ### **Category 1: Issues with Direct Solutions**
>
> ---
>
> > **Q1. Actual Specifications of the Flow and Decoder Modules**
>
> We provide the detailed specifications for the flow and decoder modules as requested.
>
> 1. Flow Module (`src/nn/flow/mixflow.py`)
>
> * Overall Structure:
>     * Number of Blocks: 10
>     * Input/Output Dimension: $D$ (latent dimension)
>
> * Composition of Each Block:
>     1.  Masked Autoregressive Flow (MAF)[1] Layer:
>         * Network: 3 masked linear layers.
>         * Dimensions: $[D, D \times \text{flow\_size}, D \times \text{flow\_size}, 2D]$
>         * Activation: `LeakyReLU`
>         * Scale Constraint: Enabled (`constrain_scale: True`)
>     2.  Batch Normalization (`BatchNorm`) Layer:
>         * Dimension: $D$
>         * Momentum: 0.5
>
> All user-configurable hyperparameters are detailed in Appendix B. Other parameters use the default values from the source code.
>
> 2. Decoder Module (`src/nn/output/decoder.py`)
>
> The decoder models the output distribution using a multivariate normal distribution with a conjugate prior.
> -   Output Class: `MultiNormalOutput`
> -   Parameters:
>     -   `use_prior: True`
>     -   `output1`: A linear layer mapping from latent dimension $D$ to output dimension $N$, predicting the mean $\boldsymbol{\mu}$.
>     -   `output2`: A linear layer mapping from latent dimension $D$ to $(N \times (N+1))/2$, predicting the elements of the covariance matrix $\boldsymbol{\Sigma}$.
> -   Prior: `NormalWishartPrior`
>
> 3. Prior Specification (`src/distributions/multinormal.py`)
>
> -   Class: `NormalWishartPrior`
> -   Parameters:
>     -   `evidence`: 1
>     -   `N`: Output dimension
>     -   $\boldsymbol{\chi_1}$: The prior's mean vector.
>     -   $\boldsymbol{\chi_2}$: The prior's second moment matrix (Prior Covariance + $\boldsymbol{\chi_1}\boldsymbol{\chi_1}^T$).
>
> ---
> > **Q2. Ablation Study of ESM2 Embeddings**
>
> We supplement the ESM2 embeddings ablation study here. (Metrics are in their original value without multiplying by 100)
>
> | Model              | PCC1   | PCC1_DEG | ACC1   | ACC1_DEG | SPCC   | ACC2   |
> | :----------------- | :----: | :------: | :----: | :------: | :----: | :----: |
> | Ours (UCE)         | 0.5641 | 0.5460   | 0.6305 | 0.7081   | 0.3111 | 0.2258 |
> | Ours (UCE) -5% Mask| 0.5704 | 0.5350   | 0.6361 | 0.7185   | -      | -      |
> | Ours (UCE) -10% Mask| 0.5845 | 0.5614   | 0.6369 | 0.7224   | -      | -      |
>
> Future users can easily switch between different gene embedding backbones (e.g., `scGPT`, `scFoundation`, `GNN`, `UCE`) by modifying the `backbone` argument in the configuration. This information will be added to the supplementary section.
>
> ---
>
> > **Q4. Statistical Significance**
>
> We performed one-sided paired t-tests comparing our model against baselines.
>
> **P-values from One-Sided Paired t-test (Our Model vs. Baseline)**
>
> | Baseline  | Metric | Norman   | Rpe1     | K562     |
> | :-------- | :----: | :------: | :------: | :------: |
> | GEARS-unc | ECE    | 4.08e-05 | 5.59e-05 | 3.85e-08 |
> | GEARS-Ens | ECE    | 4.10e-04 | 6.45e-04 | 6.05e-04 |
>
> ---
>
> > **Q6. Results Format**
>
> Thank you for the suggestion. In the revised manuscript, we will report all metrics (PCC, ACC, ECE, MSE, etc.) as their original decimal values, formatted to four decimal places, without multiplying by 100.

---

> ### Author Response · Authors · 2025-08-07
> **Issues Requiring Further Discussion**
>
> ### **Category 2: Issues Requiring Further Discussion**
>
> We offer clarifications and provide multiple solutions for your consideration.
>
> ---
>
> > **Q3.1 Comparison to CellFlow and Handling of Discrete Data**
>
> **Clarification on CellFlow:**
>
> We thank you for mentioning CellFlow. It is high-quality research on generating post-perturbation profiles. **However, we believe our framework and CellFlow serve different purposes and are not alternatives to each other.**
>
> 1.  **Primary Goal:** CellFlow is **a generative model** designed to generate single-cell phenotypes under complex perturbations and preserving heterogeneity. In contrast, PRESCRIBE is **an uncertainty quantification framework** that estimates the reliability of predictions for unseen perturbations by modeling their relationship to the training data.
> 2.  **Role of the Flow Module:** In CellFlow, **flow matching is used to improve the quality of the generative process**. In our model, the **flow module is used for density estimation** in the latent space, which is central to our uncertainty quantification approach.
> 3.  **Type of Uncertainty:** The uncertainty estimated by the two methods is complementary and not an alternative. CellFlow envisions its future stochastic extension as a valuable direction to model inherent randomness (data) uncertainty **in a deeper way**. Our framework, PRESCRIBE, is designed to capture **wider uncertainty sources** by quantifying both model uncertainty (density) and data uncertainty (entropy).
>
> **Clarification on Discrete Distributions:**
>
> **We agree with your point about discrete count data. Our model is, in fact, compatible with this discrete distribution.** Because the multivariate Poisson distribution also belongs to the natural exponential family, we can replace the current multivariate Gaussian output layer with a multivariate Poisson layer. This would involve swapping the Normal-Inverse-Wishart distribution for the appropriate conjugate prior. Our current implementation assumes a multivariate Gaussian distribution primarily to ensure a fair and direct comparison with our baselines.
>
> Based on this, we propose the following solutions:
>
> * **Solution 1:** We will **explicitly mention the modeling of discrete count data as an important direction** for our future work in the discussion section.
>
> * **Solution 2:** We will **expand the related work section** to delineate better PRESCRIBE from generative frameworks like CellFlow and CPA, emphasizing that their primary goal is disentangling and modeling complex effects, whereas ours is quantifying prediction uncertainty.
>
> > **Q3.2 Additional Uncertainty Metrics**
>
> We agree that a comprehensive evaluation of uncertainty is critical. However, for regression tasks, the set of established uncertainty metrics is less extensive than for classification. We currently report SPCC, ACC2, and ECE, and provide calibration plots, which are standard metrics referenced by others [2][3].
>
> If these are insufficient, **we would be grateful for any recommendations on other specific metrics you would find more informative for this task.**
>
> > **Q5. Underestimated Correlation and E-distance Estimation**
>
> We clarify that the low SPCC is caused by the poor quality of  **ground truth E**, not by a flaw in our E-distance estimation. The original ground truth E was "noisy" because, to include all perturbation types, it had to rely on many sparse perturbations where true E cannot be reliably measured.
>
> Table 1(b) validates this. When a more reliable ground truth E by calculated with high cell counts. While our **estimated E-distance score remains the same, its correlation (SPCC) with this higher-quality ground truth E is significantly stronger**. For clarity, we have copied both Table 1(a) and 1(b) below.
>
> Impact of Ground Truth Quality (via Cell Count) on SPCC
>
> Norman Dataset
> | Min. Cells for E-distance Calc.  | SPCC |
> | :--- | :--: |
> | 54 (Tabel 1(a))| 25.81 |
> | 200 (Tabel 1(b))| 33.33 |
> | 500 (Tabel 1(b))| 50.00 |
>
> Rpe1 Dataset
> | Min. Cells for E-distance Calc.  | SPCC |
> | :--- | :--: |
> | 2 (Tabel 1(a))| 12.18 |
> | 300 | 53.92 |
> | 400 | 61.90 |
> | 500 | 64.85 |
>
> K562 Dataset
> | Min. Cells for E-distance Calc.  | SPCC |
> | :--- | :--: |
> | 5 (Tabel 1(a))| 24.74 |
> | 300 | 50.32 |
> | 400 | 65.85 |
> | 500 | 70.90 |
>
> This demonstrates that **our E-estimation is actually effective, and its performance was previously masked by a noisy ground truth.**
>
> ---
>
> We hope our responses and new results have addressed your concerns in Category 1. For the issues in Category 2, we have proposed several solutions and would greatly appreciate your feedback.
>
> Sincerely,
>
> The Authors
>
> [1] Papamakarios et al. "Masked Autoregressive Flow for Density Estimation." NeurlPS (2018)
>
> [2] Sun, E.D. et al. TISSUE: uncertainty-calibrated prediction of single-cell spatial transcriptomics improves downstream analyses. Nat Methods (2024)
>
> [3] Rosen et al. "Kermut: Composite kernel regression for protein variant effects." NeurlPS (2024)

---

### Official Review · Reviewer_tFuR · 2025-07-02

**Clarity:** 2
**Significance:** 3
**Originality:** 3
**Rating:** 5
**Confidence:** 3

**Summary:**

This paper presents **PRESCRIBE**, a model for predicting single-cell gene expression responses to genetic perturbations while estimating uncertainty. It extends the Natural Posterior Network to a multivariate setting and introduces a pseudo E-distance metric that approximates confidence without requiring post-perturbation data. The model is evaluated on standard benchmarks, demonstrating well-calibrated uncertainty estimates and improved prediction accuracy, especially when low-confidence outputs are filtered.

**Questions:**

**Q1**: Given the use of multivariate NatPN with normalizing flows, how does PRESCRIBE scale with increasing gene dimension or perturbation complexity? What are the memory/runtime bottlenecks?

**Q2**: All benchmark datasets seem to share a common structure and origin. Did you test generalization on held-out cell types or across studies to evaluate transfer performance?

**Q3**: In the filtering-based accuracy gains,what are the characteristics of the discarded predictions?

**Ethical Concerns:**

["NO or VERY MINOR ethics concerns only"]

**Final Justification:**

The rebuttal addressed my initial concerns and included additional experiments that further support the proposed method. Critical points raised by Reviewer UDmR, such as the lack of ablation studies, were addressed with concrete results. With these new results and clarifications, and the commitment to include them in the camera-ready version, the paper demonstrates a strong contribution to the literature. I therefore maintain my positive score.

**Limitations:**

yes

**Paper Formatting Concerns:**

None observed.

**Quality:**

4

**Strengths And Weaknesses:**

## Strengths
- **Practical relevance**: The authors address a key limitation in current single-cell perturbation methods; the lack of uncertainty estimation.
- **Theoretical framework**: The derivation of the multivariate NatPN and its use of the Normal-Inverse Wishart distribution is a non-trivial extension of evidential learning approaches and seems appropriate for the single-cell perturbation task. The mathematical derivations are furthermore clearly explained.
- **Comprehensive empirical evaluation**: The authors conduct thorough experimentation on three benchmarks, comparing to both methods without uncertainty estimation, and with. Moreover, the methods are evaluated on a range of relevant metrics.

## Weaknesses
- **Accessibility**: The paper is technically very dense, with advanced Bayesian machinery. This may limit accessibility to a broad ML and computational biology audience.
- **Limited real-world validation**: The authors do not explore biological insights or interpretation of the uncertainty scores (*e.g.*, which perturbations are hard to predict and why). This is a missed opportunity to explore the model's utility beyond benchmarks.
- **Lack of standard calibration metrics**: While Spearman correlation is used between prediction accuracy and confidence, no standard calibration metrics are utilized (*e.g.*, Brier score or ECE).
- **Scalability**: The multivariate extension of NatPN with flows and Bayesian updates may introduce significant computational complexity, especially as the number of genes (dimensions) and perturbations increases. This isn’t deeply analyzed.

---

> ### Author Rebuttal · Authors · 2025-07-31
>
> Dear Reviewer tFuR,
>
> Thank you for your insightful comments and valuable suggestions. Our point-by-point responses are detailed below.
>
> **Metrics and Their Descriptions**
>
> | Metric | Purpose | Description |
> | :--- | :--- | :--- |
> | PCC1 | Performance | The Pearson Correlation Coefficient between predicted and true log fold changes for genes under a specific perturbation. |
> | ACC1 | Performance | The accuracy of predicting the sign (i.e., up- or down-regulation) of the log fold change. |
> | ACC2 | Calibration | The accuracy of using our uncertainty metric to classify predictions into quintiles of final quality (PCC1). |
> | SPCC | Calibration | The Spearman Rank Correlation between our proposed uncertainty metric and the final prediction quality (PCC1). |
> | ECE | Calibration | Expected Calibration Error. Measures the discrepancy between the model's predictive confidence and its actual accuracy. Lower values indicate better calibration. |
>
> ---
>
> > **Q1. Scalability: Given the use of multivariate NatPN with normalizing flows, how does PRESCRIBE scale with increasing gene dimension or perturbation complexity? What are the memory/runtime bottlenecks?**
>
> We benchmarked PRESCRIBE's runtime while varying the model's latent dimension, $N$. The results below show that the average batch time scales in a near-linear fashion.
>
> | Latent Dim. | Avg. Time / Batch (s) | Max Epochs to Converge |
> | :--- | :---: | :---: |
> | N | 261.53 | 15 |
> | 2N | 268.86 | 28 |
> | 3N | 272.31 | 32 |
> | 4N | 275.16 | 42 |
> | 5N | 340.66 | 50 |
>
> ---
>
> > **Q2. All benchmark datasets seem to share a common structure and origin. Did you test generalization on held-out cell types or across studies to evaluate transfer performance?**
>
>  To supplement our standard benchmarks, we performed a new cross-cell-type generalization experiment.
>
> **Experimental Design:** We used the dataset from Nadig et al. (2025), which contains CRISPR screens in Jurkat and HepG2 cells. We created a transfer task where the model was trained on all perturbations in HepG2 and a subset of perturbations in Jurkat. It was then tested on the held-out perturbations in the Jurkat cell line. The model uses a shared encoder with cell-type-specific flow and decoder modules.
>
> **Results & Interpretation:**
> As the table below shows, PRESCRIBE's uncertainty metric remains well-calibrated (SPCC > 0.38) in this transfer task. And by filtering out the 5-10% of predictions with the highest uncertainty, the performance on the remaining predictions consistently improves.
>
> | Model | PCC1 | PCC1_DEG | ACC1 | ACC1_DEG | SPCC | ACC2 |
> | :--- | :---: | :---: | :---: | :---: | :---: | :---: |
> | Ours (PRESCRIBE) | 30.77 | 33.16 | 58.88 | 71.32 | 38.45 | 26.32 |
> | Ours-5%| 31.94 | 34.10 | 59.23 | 71.45 | - | - |
> | Ours-10%| 33.12 | 34.68 | 59.50 | 71.53 | - | - |
>
> ---
>
> > **Q3 / W2. In the filtering-based accuracy gains, what are the characteristics of the discarded predictions?**
>
> Low confidence metric our model predicted induced from two uncertainty sources:
>
> For seen or familiar perturbations, low confidence means either high prediction variance (like GEARS) (learnt by entropy) or low effect magnitude (supervised by ranking loss). (Fig.1 & Table 1).
>
>  For unseen perturbations,  we expected the flow module to generate low evidence or likelihood so that it constructs a common feature for those epistemic uncertainties (Fig.3(a)) rather than an arbitrary prediction.
>
>
> > **W1. The paper is technically very dense... This may limit accessibility...**
>
> Thank you for this feedback. We will revise the introduction and methods to improve accessibility. We will clarify the core contributions for different audiences:
>
> * **For the Machine Learning Audience:** They can ignore E-distance and prior settings to directly use a multivariate nature posterior network, which mainly focuses on estimating variance and epistemic uncertainties.
> * **For the Computational Biology Audience:** We will highlight the practical utility, thus we consider the magnitude of the effect by using the E-distance concept.
>
> > **W3. While Spearman correlation is used... no standard calibration metrics are utilized.**
>
> We agree and have now benchmarked our model using Expected Calibration Error (ECE), a standard metric for calibrated regression (Kuleshov et al., ICML 2018). Lower ECE values indicate better calibration.
>
> The table below reports the ECE (multiplied by 100) for PRESCRIBE and baselines.
>
> | Model | Norman | Rep1 | K562 |
> | :--- | :---: | :---: | :---: |
> | Ours | 30.59 | 50.38 | 41.47 |
> | GEARS-unc | 32.45 | 63.96 | 64.76 |
> | GEARS-Ens | 38.63 | 53.85 | 43.07 |
>
> We will add the formal definition of ECE and a detailed discussion of these results to the revised manuscript.

---

> > ### Author Response · Authors · 2025-08-05
> > **Look forward to your feedback**
> >
> > Dear Reviewer tFuR,
> >
> > Thank you for your insightful and constructive review. **Your recommendation to use the Expected Calibration Error (ECE) is particularly valuable. Incorporating this metric has allowed us to perform a more standard uncertainty calibration for this new task, which has significantly improved our validation.** If you have further questions or need additional clarification, we would be happy to discuss them. Thank you again for your time and effort in reviewing our manuscript. Your feedback has been instrumental in improving our research.
> >
> > Best,
> >
> > The Authors

---

> > ### Comment · Reviewer_tFuR · 2025-08-05
> >
> > I thank the authors for carefully addressing all of my concerns in the rebuttal and for running additional experiments, which further strengthen the promise of the proposed method.
> >
> > I also appreciate the critical concerns raised by Reviewer UDmR, particularly regarding the lack of ablation studies. I believe the authors have responded adequately to these points in the rebuttal by providing concrete results.
> >
> > Given the new results and clarifications provided, and the authors' commitment to including them in the camera-ready version, I believe this paper makes a strong contribution to the literature. I therefore maintain my positive score.

---

> > > ### Author Response · Authors · 2025-08-07
> > > **Letter of Appreciation**
> > >
> > > Dear Esteemed Reviewer tFuR,
> > >
> > > Thank you for your thoughtful and constructive feedback. We greatly appreciate the time and effort you have invested in reviewing our paper. **Your insights have been instrumental in enhancing the quality of our work, and we are pleased that we could address the concerns you raised.** Thank you once again for your invaluable contribution to our research.
> > >
> > > Many thanks in advance!
> > >
> > > Authors

---

### Official Review · Reviewer_UDmR · 2025-07-11

**Clarity:** 1
**Significance:** 2
**Originality:** 3
**Rating:** 4
**Confidence:** 2

**Summary:**

This paper introduces the PRESCRIBE method, which uses a natural posterior network, to provide uncertainty estimates for the effects of genetic perturbations on gene expression. This method crucially does not require ensembling or multiple forward passes. The authors justify this approach for uncertainty estimation by showing that it resembles an E-value, which has been used to establish statistical significance of perturbations in experimental data where there are sets of control and perturbed cells for each perturbation.

**Questions:**

1. How is the PCC calculated in Fig. 1? Is it across gene LFCs for a given perturbation?
2. Why should perturbation embeddings be additive in Equation 7? it seems like $f_{\alpha_{2}}$ should be a set transformer or something similar.
3. What is the intuition for setting $N_{H} = N$.
4. It is unclear what you are showing in Table 1(b). Is that the correlation with the real E-value for perturbations where that are at least N cells measured post-perturbation?
5. How does an untrained model (PRESCRIBE-Null) have such good predictive performance as is shown in Table 2?
6. Is Ours-MLP trained with the E-distance ranking loss? If not, it seems like it should be for a fair ablation.

**Ethical Concerns:**

["NO or VERY MINOR ethics concerns only"]

**Final Justification:**

I have increased my rating from a 3 to a 4 due to the author's additional experiments comparing their method to two baselines to compute uncertainty: GEARS (E-distance) and scGPT gene embeddings.

The advance seems real and the method intuitively seems compelling, but it is concerning that GEARS (E-distance) performs almost as well as their method on their nonstandard calibration metrics. And when the authors move to a more standard calibration metric (ECE), a worse version of GEARS performs almost as well as their method based on their discussion with reviewer tFuR.

If the authors sufficiently improve the paper's clarity---and during the discussion period, they did make it clear that they would at least try to---then I do still feel that this method is methodologically interesting not only for this application but potentially for others. It may not be as big an advance as originally advertised, but it still seems appropriate for a venue like NeurIPS.

**Limitations:**

Limitations are not discussed in the conclusion at this point.

**Quality:**

3

**Strengths And Weaknesses:**

- The primary weakness of this paper is its clarity. (1) Multiple terms are used before they are defined (e.g. $\chi^{\text{prior}}$ and $n^{\text{prior}}$). (2) Multiple variables are used with the same name (e.g. $n$, $N$, and $N_{H}$). (3) The methods section, instead of building up intuition, inundates the reader in equations. To be fair, I am not familiar with natural posterior networks and normalizing flows, so I might be the wrong audience, but even core ideas such as how their pseudo E-distance corresponds to the real E distance are not well explained (Figure 2V is more confusing than it is helpful). In its current incarnation, I cannot imagine how this paper would be particularly legible to people who are not already both experts on natural posterior networks and perturbational models.

- It does seem to be the case that uncertainty estimates from their method are more calibrated than for GEARS, as they show in Fig. 4. But it's not the case that they are calibrated particularly well either, since they are quite far away from the y = x line.

- It is good to see though that there predictive performance, as indicated in Table 2, is quite strong. In fact, the paper seems to undersell how strong the predictive performance is.

- Since GEARS uncertainty estimates are so bad, I think two simple additional baselines should be included to prove the utility of this method: (a) The E-value between the predictions of an existing model like GEARS and the control cells. Specifically, I am wondering if the predicted magnitudes are more reflective of the model's certainty than its variance estimates. Obviously this is not helpful for distinguishing a perturbation with actually zero effect compared to a perturbation where the model doesn't know and therefore says zero. However, for perturbations with strong effects, the model's predicted effect magnitude might be a measure of its certainty.
(b) Some measure of distance between the test set perturbations and the training set perturbations in some gene embedding space (like from scGPT).

---

> ### Author Rebuttal · Authors · 2025-07-31
>
> Dear Reviewer UDmR,
>
> Thank you for your thoughtful review and constructive comments. We have addressed each of your points below.
>
> **Metrics and Their Descriptions**
> | Metric | Usage|Description|
> |------|------|------|
> | PCC1 | Performance Evaluation| The Pearson correlation between the predicted and true log fold change under specific perturbations.|
> | ACC1 | Performance Evaluation| The directional accuracy between the predictive and true log fold change under specific perturbations.|
> | ACC2 | Caliberation Evaluation|The quintile classification accuracy between certain metric and PCC1|
> | SPCC| Caliberation Evaluation| The Spearman correlation between certain metric and PCC1|
>
> ***
>
> ### **Responses to Questions**
>
> > **Q1. How is the PCC calculated in Fig. 1? Is it across gene LFCs for a given perturbation?**
>
> **Yes.** This is a standard evaluation metric in the field. We use log-fold changes (LFCs) for two reasons:
> 1.  **Biological Relevance:** Accurately predicting the *change* induced by a perturbation is more useful since the downstream study on the perturbation effect is conducted by comparison.
> 2.  **Scoring Robustness:** Evaluating PCC directly on post-perturbation expression can lead to artificially inflated scores due to the high sparsity in single-cell data.
>
>
> > **Q2. Why should perturbation embeddings be additive in Equation 7? It seems like it should be a set transformer or something similar.**
>
> 1.  **Model Design:** Our encoder in Equation. 7 models both an additive effect (first term) and a non-linear interaction effect (second term). This architecture was adopted from prior work [1] and chosen for its effectiveness and simplicity. The encoder module is flexible, requiring only a deterministic mapping from inputs (c, x) to latent embeddings z (Appx.F. 6, Theorem 2).
>
> 2.  **Experiment with a Set Transformer:** Following your suggestion, we implemented a version of our model using a Set Transformer as the encoder. The results are presented below.
>
> | Model | PCC1 | PCC1_DEG | ACC1 | ACC1_DEG | SPCC | ACC2 |
> | :--- | :---: | :---: | :---: | :---: | :---: | :---: |
> | Ours(Set) | 57.16 | 56.15 | 63.00 | 68.33 | 12.50 | 16.13 |
> | Ours(Set)-5% | 57.40 | 56.61 | 63.31 | 68.55 | :---: |:---:  |
> | Ours(Set)-10% | 57.67 | 57.79 | 63.35 | 68.62 | :---: |:---:  |
>
> Our detailed adaptation is
>
> $\sum f_{a1}(x_i) + f_{a2}(\sum f_{a1}(x_i)) \rightarrow f_{set}(CONCAT[f_{a1}(x_1),f_{a1}(x_2),...,f_{a12}(c)])$.
>
> While this is a promising direction, the Set Transformer did not outperform our original model on this dataset, potentially due to the large data requirements for training a representative transformer.
>
> > **Q3. What is the intuition for setting $N_H=N$?**
>
> Our choice of $N_H = N$ is based on an empirical evaluation of four heuristics proposed in the NatPN paper. We tested these strategies on the Norman dataset to select the optimal setting for our model.
>
> | Model | PCC1 | PCC1_DEG | ACC1 | ACC1_DEG |SPCC | ACC2 |
> | :--- | :---: | :---: | :---: | :---: |:---: | :---: |
> | Ours(exp-half) | 14.31 | 44.47 | 50.90 | 66.94 |17.88 | 16.13 |
> | Ours(constant) | 14.31 | 44.47 | 50.90 | 66.94 |15.15 | 16.13 |
> | Ours(normal) | 50.69 | 58.02 | 61.32 | 68.32 |62.90 | 29.03 |
> | Ours(normal)-5%| 52.32 | 58.68 | 61.86 | 69.14 |:---: | :---: |
> | Ours(normal)-10%|  54.60 | 60.19 | 62.61 | 69.63 |:---: | :---: |
>
> As shown, the 'exp-half' ($N_H = e^{N/2}$) and 'constant' ($N_H = 1$) settings led to training collapse. The 'normal' setting ($N_H=e^{log(\sqrt{4\pi})N}$) underperformed the 'exp' setting ($N_H = N$) (Please refer Table 2 in the paper). We will include these analyses in the appendix.
>
> > **Q4. It is unclear what you are showing in Table 1(b).**
>
> In Table 1(b), $N$ represents the number of cells sampled from the experimental data to calculate the "ground truth" E-distance. The purpose of this table is to show that the correlation between our predicted E-distance and the ground truth improves as the ground truth becomes more reliable (i.e., as $N$ increases). This suggests that the performance reported in Table 1(a) may be underestimated, as the ground-truth E-distance calculation itself is limited by the number of available cells in the dataset.
>
> > **Q5. How does an untrained model (PRESCRIBE-Null) have such good predictive performance as is shown in Table 2?**
>
> The PRESCRIBE-Null model, while its network weights are random, makes predictions by leveraging the prior distribution, which is conditioned on the control cell profile. However, its performance is substantially lower than a more meaningful baseline like 'AverageKnown', which highlights the value of our fully trained model.
>
> > **Q6. Is Ours-MLP trained with the E-distance ranking loss? If not, it seems like it should be for a fair ablation.**
>
> Thank you for this valuable point. For a fair comparison, we re-trained the 'Ours-MLP' baseline using the same E-distance ranking loss as our whole model. The updated results are below.
>
> | Model | PCC1 | PCC1_DEG | ACC1 | ACC1_DEG | SPCC | ACC2 |
> | :--- | :---: | :---: | :---: | :---: | :---: | :---: |
> | Ours-MLP | 55.60 | 55.86 | 62.40 | 69.84 | 28.71 | 19.35|
>
> ***
>
> ### **Responses to Weaknesses**
>
> > **W1. The primary weakness of this paper is its clarity... the methods section... inundates the reader in equations...**
>
> We sincerely apologize for this and will revise the manuscript to improve clarity.
>
> (1)&(2). In the original paper, we have compiled a comprehensive notation table in Appendix A. We recognize that overloading the symbol $N$ created unnecessary confusion (Q4), and we apologize for this oversight. In the revision, we will implement the following changes: The symbol $N$ will be used exclusively to represent the latent dimension. A new, distinct symbol (e.g., $N_{cell}$) will be introduced for the number of sample cells used in the E-distance calculation.
>
> (3). We will reframe the method to clarify our method: **to create a unified confidence score** that handles multiple sources of uncertainty.
>  * **For seen or familiar perturbations**, E-distance is a powerful metric because it considers both prediction variance (like GEARS) and the predicted effect magnitude. (Fig.1 & Table 1), So identifying this case is our primary goal.
>  * **For unseen perturbations**, our NatPN-based approach is designed to map these cases to a low E-distance, which constructs a common feature for them (Fig.3(a)), which is better than arbitrary prediction.
>
>     To demonstrate this works, we supplemented an experiment that shows our model's advantage at identifying these cases (PCC<0.2 & E>0). Analysis conducted on 600+ test points across three datasets.
>
>     | % of Data Filtered by Metric | 5% | 10% | 15% | 20% |
>     | :--- | :---: | :---: | :---: | :---: |
>     | Ours Discovery Rate (%) | 17.1 | 22.9 | 25.7 | 31.4 |
>     | GEARS Discovery Rate (%)| 8.2 | 12.2 | 16.3 | 24.5 |
>
> > **W2. It does seem to be the case that uncertainty estimates from their method are more calibrated... But it's not the case that they are calibrated particularly well either...**
>
> We think that one limiting factor is the calculation of the ground-truth E-distance itself, which requires a large number of cells to be genuinely accurate, a number often unavailable in real datasets. However, the consistent trend of performance improvement upon filtering (as shown in Table 2) demonstrates that our uncertainty metric has already been put into practical use.
>
> > **W3. It is good to see though that there predictive performance... is quite strong. In fact, the paper seems to undersell how strong the predictive performance is.**
>
> Our primary focus was on validating our uncertainty quantification framework, so we emphasized the consistent trend of performance improvement upon filtering.
>
>
> > **W4. Since GEARS uncertainty estimates are so bad, I think two simple additional baselines should be included...**
>
> Thank you for suggesting these excellent baselines. We have run both experiments.
>
> **(a) GEARS Predicted Effect Magnitude as Uncertainty:** We calculated the E-distance between GEARS's predictions and the control state to use as an uncertainty score.
>
> | Uncertainty Metric | SPCC | ACC2 |
> | :--- | :---: | :---: |
> | GEARS (Original Unc.) | 37.32 | 28.12 |
> | GEARS (E-distance) | 59.82 | 46.07 |
>
> The results show this is indeed a strong baseline. We think in perturbation models (different from just a regression task). E-distance is a more comprehensive metric than variance alone. It is not a parallel concept to variance; instead, it's an inclusive one, capturing both variance (via self-distance) and effect magnitude (via pairwise-distance). The following correlations between these components and GEARS's original variance score support this:
>
> | E-distance Component | Correlation (SPCC) w/ GEARS (Original Unc.) |
> | :--- | :---: |
> | Self-distance | 10.15 |
> | Pairwise-distance | -92.38 |
>
> **(b) Embedding Similarity as Uncertainty:** We also evaluated using the cosine similarity between gene embeddings of test and training perturbations as a confidence score. We used the Rpe1 and K562 datasets, as defining similarity for the double knockouts in the Norman dataset is ambiguous.
>
> | Model (Rpe1) | SPCC | ACC2 |
> | :--- | :---: | :---: |
> | mean | 15.50 | 24.14 |
> | max | 19.83 | 22.02 |
> | median | 13.56 | 25.46 |
>
> | Model (K562) | SPCC | ACC2 |
> | :--- | :---: | :---: |
> | mean | 1.58 | 18.52 |
> | max | 10.77 | 22.96 |
> | median | 0.24 | 20.37 |
>
> This straightforward similarity-based approach does not yield a well-calibrated uncertainty metric, highlighting the need for a more sophisticated method.
>
> ***
>
> ### **Responses to Limitations**
>
> > **Limitations are not discussed in the conclusion at this point.**
>
> You can find limitations in Appendix J, where we discuss the model's reliance on the quality of perturbation embeddings and the challenges that remain in fully disentangling epistemic uncertainty.
>
> [1] SEASON COMBINATORIAL INTERVENTION PREDICTIONS WITH SALT & PEPER. ICLR (2024)

---

> > ### Author Response · Authors · 2025-08-05
> > **Looking forward to your feedback & A Follow-up Experiment on Distinguishing "Zero Effect vs. Model Uncertainty"**
> >
> > Dear Reviewer UDmR,
> >
> > As the discussion period is closing, we sincerely look forward to your feedback. We deeply appreciate your valuable time and efforts.
> >
> > **To address your concern about distinguishing "zero effect" from "model uncertainty," we further conducted a brief experiment showing how changing the setting of the prior distribution can solve this.**
> >
> > First, we identified the top 10% of perturbations with the highest uncertainty under two different prior settings: the standard control prior ($μ_0​$) and a prior using the average of the training data ($μ_{all}$​). Then, we took the intersection of the two "top 10% uncertain" sets. The results are in the table below:
> > | 10% Filtering | Averaged E-distance(vs. x) | CBL+UBASH3B | ETS2+MAPK1 | IKZF3+MAP2K6 |
> > | :--- | :--- | :--- | :--- | :--- |
> > | **Ours (x = $μ_0​$)** | 71.44 | 98.35 | 105.93 | 103.70 |
> > | **Ours (x = $μ_{all}$​)** | 38.23 | 41.30 | 54.32 | 42.86 |
> >
> > This intersection finally generated three specific perturbations in total, shown in the table. These three perturbations consistently exhibit higher E-distances regardless of the prior, which we attribute to true model uncertainty.
> >
> > Therefore, changing the prior setting provides a potential solution to distinguish between a zero effect and model uncertainty, as the former arises from the chosen prior.
> >
> > As we discuss in the conclusion, this feature allows our model to explore further the influence of various control selections on perturbation responses, thereby helping to guide optimal experimental design.(lines 277-280)
> >
> > We strive to improve the paper consistently, and it is our pleasure to have your feedback!
> >
> > Best regards,
> >
> > Authors

---

> > > ### Author Response · Authors · 2025-08-08
> > > **Looking forward to your feedback**
> > >
> > > Dear Reviewer UDmR:
> > >
> > > As the discussion period is closing, we sincerely look forward to your feedback. We deeply appreciate your valuable time and efforts. It would be very much appreciated if you could once again help review our responses and let us know if these address or partially address your concerns and if our explanations are heading in the right direction. Please also let us know if there are further questions or comments about this paper. We strive to improve the paper consistently, and it is our pleasure to have your feedback!
> > >
> > > Best regards,
> > >
> > > Authors

---

> > ### Comment · Reviewer_UDmR · 2025-08-08
> > **Response to Author Rebuttal**
> >
> > Thank you for answering my questions, conducting the follow-up experiments, and being willing to focus on clarity in the revisions.
> >
> > Could you please clarify what your method's results are for the additional experiments in which you (a) trained Ours-MLP with a ranking loss and (b) used pseudo E-distance from GEARS predictions as a measure of uncertainty. I am having a hard time finding the table or figure from your paper with the numbers to compare against.
> >
> > As an aside, in the final version of the paper I would encourage you not to label metrics as PCC_{1} and PCC_{2}. While the shorthand saves space, these are quite confusing. Maybe, PCC_{1} can become $r_{\text{pred}, \text{truth}}$ and PCC_{2} can become $r_{\text{perf}, \text{uncertainty}}$. Additionally some metrics were not well defined in the original paper. For example, I only now understand what ACC_{2} is given the metrics table in your response.

---

> > > ### Author Response · Authors · 2025-08-09
> > > **Clarifications on Results and Updated Metric Notation**
> > >
> > > Dear Reviewer UDmR,
> > >
> > > Thank you for your constructive feedback and for the opportunity to clarify our results. Our responses are detailed below.
> > >
> > > ---
> > >
> > > ### **1. Clarification on Results**
> > >
> > > We provide the complete table of results for the additional experiments. This includes performance for (a) Ours-MLP with a ranking loss and (b) GEARS with pseudo E-distance.
> > >
> > > **Table 1: Performance on the Norman2019 dataset**
> > > | Method | $r_{\text{pred}, \text{truth}}$ |$r^{\text{DEG}}_{\text{pred}, \text{truth}}$ | $\text{ACC}_{\text{pred}, \text{truth}}$ | $\text{ACC}^{\text{DEG}}_{\text{pred}, \text{truth}}$ | $r^{s}_{\text{perf}, \text{conf}}$ | $\text{ACC}_{\text{perf}, \text{conf}}$ |
> > > | :--- | :---: | :---: | :---: | :---: | :---: | :---: |
> > > | Ours-MLP (a) | 55.60 | 55.86 | 62.40 | 69.84 | 28.71 | 19.35 |
> > > | GEARS (Original Unc.) | 45.30 | 63.19 | 29.09 | 69.06 | 37.32 | 28.12 |
> > > | GEARS (E-distance) (b) | 45.30 | 63.19 | 29.09 | 69.06 | 59.82 | 46.07 |
> > > | **Ours (Proposed)** | **58.38** | **64.44** | **63.24** | **74.68** | **67.19** | **51.61** |
> > >
> > > *Kindly note, the corresponding SPCC $r\^{s}\_{\text{perf}, \text{conf}}$ results of our model can be found in Figure 4(a) of the main paper.*
> > >
> > > (1).  Ours-MLP (a): The MLP-based model exhibits weaker calibration performance. We attribute this to the difficulty of using an MLP's latent space to differentiate between familiar (high-density) and unfamiliar (low-density) inputs.
> > >
> > > (2).  GEARS (E-distance) (b): Applying E-distance improves the calibration of GEARS, likely because it captures both covariance and the perturbation's effect intensity. However, this model is still outperformed by our proposed method, which is designed to account for model uncertainty. We provide a solution that distinguishes these effects by adjusting the model's prior (see our previous response for details).
> > >
> > > ---
> > >
> > > ### **2. Updated Metric Notation**
> > >
> > > We agree with your suggestion and thank you for the recommendation. Using descriptive names is more intuitive, and we have updated our metric definitions as follows:
> > >
> > > | Metric | Usage | Description |
> > > | :--- | :--- | :--- |
> > > | $r_{\text{pred}, \text{truth}}$ | Performance | The Pearson correlation between the *pred*icted and *true* log-fold changes. |
> > > | $\text{ACC}_{\text{pred}, \text{truth}}$ | Performance | The directional accuracy between the *pred*icted and *true* log-fold changes. |
> > > | $r^{s}_{\text{perf}, \text{conf}}$ | Calibration | The Spearman correlation between a model's *perf*ormance ($r_{\text{pred}, \text{truth}}$) and its *conf*idence score. |
> > > | $\text{ACC}_{\text{perf}, \text{conf}}$ | Calibration | The quintile classification accuracy, measuring how well a *conf*idence score predicts *perf*ormance ($r_{\text{pred}, \text{truth}}$) quintiles. |
> > >
> > > #### Other Terms
> > >
> > > * DEG: Top 20 differentially expressed genes.
> > > * $\mu_0$: The unperturbed cellular state.
> > > * $\mu_\text{all}$: The set of all cellular states in the training set.
> > > * $\varepsilon_{\text{perf}, \text{conf}}$: Expected Calibration Error (ECE). This is a standard calibration metric (Kuleshov et al., ICML 2018), as recommended by Reviewer tFuR.
> > >
> > > For your convenience, we are re-sharing the results of the cosine similarity baseline. In this analysis, `mean`, `max`, and `median` represent statistics calculated from the cosine similarities between a test perturbation and all training perturbations.
> > >
> > > **Table 2: Cosine-Similarity Baseline (Rpe1)**
> > > | Metric | $r^{s}_{\text{perf}, \text{conf}}$ | $\text{ACC}_{\text{perf}, \text{conf}}$ |
> > > | :---   | :---: | :---: |
> > > | mean   | 15.50 | 24.14 |
> > > | max    | 19.83 | 22.02 |
> > > | median | 13.56 | 25.46 |
> > > | **Ours** | **28.69** | **27.32** |
> > >
> > > **Table 3: Cosine-Similarity Baseline (K562)**
> > > | Metric | $r^{s}_{\text{perf}, \text{conf}}$ | $\text{ACC}_{\text{perf}, \text{conf}}$ |
> > > | :---   | :---: | :---: |
> > > | mean   | 1.58  | 18.52 |
> > > | max    | 10.77 | 22.96 |
> > > | median | 0.24  | 20.37 |
> > > | **Ours** | **12.43** | **24.81** |
> > >
> > >
> > > Thank you again for your valuable feedback. We hope these clarifications and revisions address your concerns and have improved the quality of our paper.

---

### Official Review · Reviewer_HczP · 2025-07-21

**Clarity:** 3
**Significance:** 2
**Originality:** 3
**Rating:** 5
**Confidence:** 3

**Summary:**

In order to address the issue of uncertainty estimation in modeling multidimensional perturbation datasets like those containing  transcriptomic readouts, the authors propose PRESCRIBE, an extension of the Natural Posterior Network (NatPN) accommodating multivariate normally distributed data. The authors additionally interpret the NatPN confidence score as "pseudo" energy distance (or E-distance), add a loss term to ensure that this interpretation holds, and attempt to show that the pseudo E-distance are well-calibrated with predictive performance.

**Questions:**

1. Why did you focus on extending NatPN with energy distances? What motivated this? The paper suggests that it is because of an apparent similarity between E-distance and the NatPN confidence, but that is not obvious.
1. The results of Table 3 are confusing - why does random filtering of 10% outperform that of 5% and match no filtering?
1. Why is it important to keep the evidence in the range [N, 2N]? Preventing the t-distribution from resembling a Gaussian is mentioned, but I don't understand the negative implication that resembling a Gaussian would have.
1. Why the form of the encoder mentioned in section 3.3? Is it appropriate to assume additive effects? The authors cite Gaudelet et a.l. (2024) for this result, but that paper is clear that the approach is more or less appropriate depending on the interaction. An analysis of the E-distance by interaction type would be an insightful extension of the analysis of that paper.

**Ethical Concerns:**

["NO or VERY MINOR ethics concerns only"]

**Final Justification:**

Authors thoroughly addressed my concerns.

**Limitations:**

1. The recent paper [1] explores the performance issues of many of the baseline models and datasets used in this paper, and recommends new metrics that better demonstrate predictive performance. This paper should at least acknowledge this as a limitation, and should very likely redo some of the analyses with the suggested metrics.

[1] Mejia, Gabriel M., et al. "Diversity by Design: Addressing Mode Collapse Improves scRNA-seq Perturbation Modeling on Well-Calibrated Metrics." arXiv preprint arXiv:2506.22641 (2025).

**Quality:**

3

**Strengths And Weaknesses:**

Strengths:
1. Valid extension of NatPN to multivariate data and concepts of energy distance
1. Extensive set of comparisons and ablations intending to establish the claims of predictive performance and well-calibrated uncertainty estimation
1. The method accommodates combinatorial perturbations

Weaknesses:
1. The introduction of energy distance is poorly motivated
1. There is application to only 3 datasets of one data and perturbation type, when this could have been fruitfully applied to many related datasets with different perturbation types, and even a toy datasets to illustrate the versatility of PRESCRIBE
1. Following on the previous point, claims of PRESCRIBE being well-calibrated are limited to the narrow set of empirical results included in the paper. Results like Figure 3b seem cherry-picked.
1. Model choices that could have been insightfully explored through ablation but were not: scGPT embeddings for gene perturbation, encoder architecture, PCA dimensions
1. There is some confusion about the $\lambda$s in equation 9 and the derivations in Appendix F. I think Section 3.4 should include a $\lambda_i$ in front of each $\mathcal{L}_i$, and the notation in Section 4.3 updated accordingly.

---

> ### Author Rebuttal · Authors · 2025-07-31
>
> Dear Reviewer HczP,
>
> Thank you for all your helpful comments. The following are our responses, following the order of questions, weaknesses, and limitations.
>
> > **Q1/W1.Why did you focus on extending NatPN with energy distances?**
>
> Our motivation was to construct trustworthy perturbation models, using NatPN as it is efficient, flexible, and can capture both model and data uncertainty. We used the E-distance because previous perturbation-related research[1] suggests that the perturbation E-distance can reflect data quality, and we further find that it can efficiently discover unsatisfactory results, so we try to combine them.
>
> We admit that the link with pairwise distance with evidence causes some confusion, as we want to incorporate them together. Actually, for broad ML users, they don't need to consider E-distance and L3; they can just use our extended multivariate NatPN. For perturbation models, we prioritize the utility, so we introduce pairwise distance and E-distance.
>
> > **Q2.The results of Table 3 are confusing - why random..**
>
> This table aims to ablate the impact of filtering. Performance can be significantly enhanced by filtering out the few worst predictions, such as those with negative PCCs in Fig. 1(a). Therefore, as the number of filtered items increases, there is a higher probability of removing the worst predictions, which in turn increases performance. Thus, the stable trend is more significant.
>
> > **Q3.Why is it important to keep the evidence in the range [N, 2N]?**
>
> When the t-distribution degenerates into a Gaussian, the model behaves like GEARS, which only mimics variance. This means it tends to interpolate and underestimate epistemic uncertainty in missing or sparse areas of the perturbation space, e.g., for new perturbation inputs.
>
> > **Q4.Why the form of the encoder mentioned in section 3.3? Is it appropriate to assume additive effects?**
>
> 1.  Model Design: The encoder module actually is flexible, requiring only a deterministic mapping from inputs (c, x) to latent embeddings z (Appx.F. 6, Theorem 2), so we take a simple and efficient one.
>
> 2. We do not find that the E-distance has a significant relationship with the interaction type. Here are the mean E-distances for several common interaction types analyzed on Norman.
>
> | Interaction | E-distance |
> | :--- | :--- |
> | Additive | 62.02 |
> | Epistasis | 66.65 |
> | Neomorph | 19.32 |
> | Redundant | 176.76 |
> | Suppressive | 43.86 |
> | Synergy | 80.42 |
>
> > **W2.There is an application to only three datasets of one data and perturbation type, when ..**
>
> Actually, our baselines, like scGPT and GEARS, all use these three typical datasets, so we followed the same convention. As for transfer performance, we consider two scenarios.
>
> The first is perturbation transfer, such as predicting the effects of new genetic or drug perturbations; the adaptation is simple, replacing $x_i$ with the new perturbation embeddings.
>
> The second is cell type or basal state transfer, which is trickier since different cell lines have very different systems. We conducted a trial where we held out some perturbations in one cell type and used another known cell type to predict them. The design is that two cell types share the same encoder but have different flow and decoder modules. We used the dataset from Nadig et al. (2025), which contains CRISPR screens in Jurkat and HepG2 cells as an example.
>
> **Results & Interpretation:**
> As the table below shows, PRESCRIBE's uncertainty metric remains well-calibrated (SPCC > 0.38) in this transfer task. And by filtering out the 5-10% of predictions with the highest uncertainty, the performance on the remaining predictions consistently improves.
>
> | Model | PCC1 | PCC1_DEG | ACC1 | ACC1_DEG | SPCC | ACC2 |
> | :--- | :---: | :---: | :---: | :---: | :---: | :---: |
> | Ours (PRESCRIBE) | 30.77 | 33.16 | 58.88 | 71.32 | 38.45 | 26.32 |
> | Ours-5% Filtered | 31.94 | 34.10 | 59.23 | 71.45 | - | - |
> | Ours-10% Filtered | 33.12 | 34.68 | 59.50 | 71.53 | - | - |
>
> > **W3.Following on the previous point, claims of PRESCRIBE being well-calibrated are limited to the narrow set of empirical results included in the paper. Results like Figure 3b seem cherry-picked.**
>
> (1) We think that one limiting factor is the calculation of the ground-truth E-distance itself, which requires a large number of cells to be genuinely accurate, a number often unavailable in real datasets. However, the consistent trend of performance improvement upon filtering (as shown in Table 2) demonstrates that our uncertainty metric has already been put into practical use.
>
> (2) Additionally, we updated more standard calibration metrics using Expected Calibration Error (ECE), a standard metric for calibrated regression (Kuleshov et al., ICML 2018). Lower ECE values indicate better calibration.
>
> The table below reports the ECE (multiplied by 100) for PRESCRIBE and baselines.
>
> | Model | Norman | Rep1 | K562 |
> | :--- | :---: | :---: | :---: |
> | Ours | 30.59 | 50.38 | 41.47 |
> | GEARS-unc | 32.45 | 63.96 | 64.76 |
> | GEARS-Ens | 38.63 | 53.85 | 43.07 |
>
> We will add the formal definition of ECE and a detailed discussion of these results to the revised manuscript.
>
> > **W4.Model choices that could have been insightful..**
>
> We provide ablation studies below.
> | Model | PCC1 | PCC1_DEG | ACC1 | ACC1_DEG |SPCC | ACC2 |
> | :--- | :--- | :--- | :--- | :--- |:---: |:---:  |
> | Ours (scFoundation) | 57.79 | 56.86 | 63.35 | 70.34 |37.02 | 22.58|
> | Ours (scFoundation)-5% | 59.68 | 58.36 | 63.75 | 70.56 |:---: |:--:|
> | Ours (scFoundation)-10% | 61.46 | 58.63 | 64.47 | 70.65 |:---: |:---:  |
> | Ours (GNN) | 14.31 | 44.47 | 50.90 | 66.94 |15.15 | 16.13 |
> | Ours (GNN)-5% | 14.58 | 45.29 | 50.87 | 66.90 |:---: |:---:  |
> | Ours (GNN)-10% | 14.76 | 45.62 | 51.03 | 67.04 |:---: |:---:|
> | Ours(Set) | 57.16 | 56.15 | 63.00 | 68.33 | 12.50 | 16.13 |
> | Ours(Set)-5% | 57.40 | 56.61 | 63.31 | 68.55 | :---: |:---:  |
> | Ours(Set)-10% | 57.67 | 57.79 | 63.35 | 68.62 | :---: |:---:  |
> | Ours | 58.38 | 64.44 | 63.24 | 74.68 |67.19|38.71|
> | Ours-5% | 61.58 | 66.36 | 64.08 | 75.69 |:---: |:---:|
> | Ours-10% | 64.32 | 68.61 | 64.73 | 75.93 |:---: |:---:|
>
> > **W5.There is some confusion about the $\lambda$ s in equation 9 ...**
>
> (1) We do not add a $\lambda$ coefficient for $L_1$ because it is the central optimization function, and we wanted to reduce the hyperparameter tuning burden; other parameters are all smaller than 1.
>
> (2) We think the coefficient does not affect this core design: when $\partial L_1/\partial p_v$ approaches zero, $\lambda_3 \partial L_4/\partial p_v \neq 0$ when $\lambda_3 \neq 0$.
>
> > **Limitation The recent paper [1] explores the performance issues of many of the baseline models and datasets used in this paper.**
>
> After reading this paper, we agree that the reference frame is essential in perturbation models. We think your concern may be that the E-distance is only correlated with the old metrics. Therefore, we conducted a quick preliminary study where we used $\mu_{all}$ to recalculate the GEARS metric and obtained the following results:
>
> | Model | PCC(new) |
> | :--- | :--- |
> | unc | 23.42 |
> | E-distance (new)| 56.05 |
> | E-distance (old)| 55.57 |
>
> We find that although the new E-distance is more correlated with the new PCC, it does not dominate the strong E-distance signal with PCC(new), as the old E-distance is still highly correlated with the new PCC.
>
> And since the paper was published very recently, we do not have time to update all our research, but would like to include it in the future.
>
> [1] Peidli, S., Green, T.D., Shen, C. et al. scPerturb: harmonized single-cell perturbation data. Nat Methods.(2024)

---

> ### Author Response · Authors · 2025-08-08
> **Looking forward to your feedback**
>
> Dear Reviewer HczP:
>
> **We've updated our general response with a more detailed and clear explanation of our core idea and modules, based on your feedback.** We would be grateful if you could review our responses one last time before the discussion period ends and let us know if we've addressed your concerns. Your guidance has been extremely helpful.
>
> Best regards,
>
> Authors

---

> ### Author Response · Authors · 2025-08-09
>
> Dear Reviewer HczP,
>
> We understand that chasing down your reply is not our job, and we do not intend to add any pressure on your busy schedule. However, as we are getting closer to the end of the discussion phase, we would really appreciate it if you could be so kind as to let us know if we have properly addressed your comments and questions in the rebuttal and if anything can be further clarified.
>
> Many thanks in advance!
>
> Authors

---

### Official Review · Reviewer_YtU5 · 2025-07-23

**Clarity:** 3
**Significance:** 3
**Originality:** 4
**Rating:** 5
**Confidence:** 3

**Summary:**

This paper presents a method called PRESCRIBE that predicts how single cells respond to perturbations. For each prediction, it also provides an uncertainty score, which it calls a "pseudo E-distance".

The model builds on the Natural Posterior Network (NatPN) but adapts it for this problem in a few key ways. It extends the network to a multivariate case using the Normal-Inverse-Wishart distribution. It also adds two new loss terms: one to make its uncertainty score mimic the ranking of true E-distances from the training data, and another to help with training by preventing zero gradients.

The paper tests PRESCRIBE's prediction accuracy and the quality of its uncertainty scores on three benchmark datasets and includes ablation studies to back up its design choices.

**Questions:**

- The author(s) initially present experimental results for all the datasets and then focus the rest of the exploration on the Norman dataset. What is the reason for focusing on the Norman dataset for the later analyses? Are the results robust across all three datasets? Similar experimental results for the other two datasets (e.g. in the Appendix) would be helpful here and could improve the clarity and significance of this work. If this is not feasible, an explanation for why this analysis is specific to the Norman dataset is necessary for transparency.

- In the presentation of the method in 3.1, the author(s) fix the proportion between $n$, $\kappa$ and $\nu$ (Eq. 4). Could the authors provide the theoretical or empirical justification for this specific modeling choice? A brief ablation study showing that the model is robust to this choice, or that this choice is optimal, would strengthen the clarity of the model design choices here.

- How robust the model is to misspecification of its Normal-Inverse-Wishart distribution? It may be nice to know how strong the regularization of the $\mathcal{L}_3$ term is in handling any model misspecification here since it encourages the pseudo E-distance derived from the Normal-Inverse-Wishart prior to match the training data E-distance. A discussion of its robustness or a simple experiment on synthetic data with a known non-Gaussian ground truth (but still reasonable for single cell settings) to demonstrate that not only the predictions but also the uncertainty scores are robust would clarify on the limitations and strengths of this paper.

**Ethical Concerns:**

["NO or VERY MINOR ethics concerns only"]

**Final Justification:**

I already gave the paper an accept rating and the authors were able to clarify on my concerns.

**Limitations:**

Yes

**Quality:**

4

**Strengths And Weaknesses:**

**Quality:** Overall, I believe this is a high-quality paper. It addresses the critical need for reliable uncertainty quantification in single-cell response prediction. The authors present a convincing framework, PRESCRIBE, that is technically solid. The quality of the work is supported by extensive experiments on three benchmark datasets, as well as thorough ablation studies that justify the model's core components.

**Clarity:** The paper is clearly written and well-structured, with helpful figures illustrating the method. However, the clarity could be improved by providing more explicit justification for certain design choices (e.g. see Questions).

**Significance + Originality:** While my expertise is not in single-cell perturbation modeling, I’ve assessed the work based on the context provided in the paper. Within that frame, the work seems to be an original and significant contribution. The authors propose a novel method that balances prediction accuracy with uncertainty quantification, supported by ample theoretical motivation. They introduce well-motivated notions of uncertainty under the Normal-Inverse-Wishart model – which has been used in previous prediction models in this setting. The work's significance is clear as it has the potential to make in-silico screening more reliable by allowing researchers to know when to trust a specific prediction—a critical need for applications like drug discovery and the development of genetic therapies.

---

> ### Author Rebuttal · Authors · 2025-07-31
>
> Dear Reviewer YtU5,
>
> Thank you for all your helpful comments. The following are our responses.
>
> ### **Metrics and Their Descriptions**
>
> | Metric | Purpose | Description |
> | :--- | :--- | :--- |
> | **PCC1** | Performance | The Pearson Correlation Coefficient between predicted and true log fold changes for genes under a specific perturbation. |
> | **ACC1** | Performance | The accuracy of predicting the sign (i.e., up- or down-regulation) of the log fold change. |
> | **ACC2** | Calibration | The accuracy of using our uncertainty metric to classify predictions into quintiles of final quality (PCC1). |
> | **SPCC** | Calibration | The Spearman Rank Correlation between our proposed uncertainty metric and the final prediction quality (PCC1). |
>
> > **Q1. The author(s) initially present experimental results for all the datasets and then focus the rest of the exploration on the Norman dataset. What is the reason for focusing on the Norman dataset for the later analyses? Are the results robust across all three datasets? Similar experimental results for the other two datasets (e.g., in the Appendix) would be helpful here and could improve the clarity and significance of this work. If this is not feasible, an explanation for why this analysis is specific to the Norman dataset is necessary for transparency.**
>
> (1) We conducted experiments across all datasets, with the exception of the N-increasing analysis, generalization analysis, and ablation study. The following are the reasons and supplement experiments.
>
> (2) We supplement the N-increasing analysis for all datasets here.
>
> | Cell Count (Rpe1) | SPCC |
> | :--- | :--- |
> | 300 | 53.92 |
> | 400 | 61.90 |
> | 500 | 64.85 |
>
> | Cell Count (k562) | SPCC |
> | :--- | :--- |
> | 300 | 50.32 |
> | 400 | 65.85 |
> | 500 | 70.90 |
>
> (3) For the generalization analysis, we focused on the Norman dataset because it involves double knock-outs, which allows us to mimic different degrees of generalization. The other datasets involve single knock-outs, for which the degree of generalization is harder to define. Therefore, for those datasets, we provide the average training and test set confidence scores instead.
>
> | Model (Rpe1) | Seen | Unseen |
> | :--- | :--- | :--- |
> | GEARS (uncertainty)| 29.47 | 25.97 |
> | Ours  (confidence)| 15.70 | 8.76 |
>
> | Model (k562) | Seen | Unseen |
> | :--- | :--- | :--- |
> | GEARS (uncertainty)| 36.24 | 31.61 |
> | Ours (confidence)| 18.10 | 14.35 |
>
> (4) For the ablation study, constrained by time, we only provide an ablation on the loss function here.
>
> | Model (Rpe1) | PCC1 | ACC1 | SPCC |
> | :--- | :--- | :--- | :--- |
> | Ours | 59.18 | 67.36 | 28.69 |
> | w/o L2 | 58.86 | 63.72 | -1.49 |
> | w/o L3 | 8.50 | 51.83 | 5.75 |
> | w/o L4 | 8.50 | 51.83 | 5.75 |
> | NO-INFO | 58.89 | 68.72 | -5.00 |
>
> | Model (K562) | PCC1 | ACC1 | SPCC |
> | :--- | :--- | :--- | :--- |
> | Ours | 36.20 | 60.27 | 12.43 |
> | w/o L2 | 35.57 | 60.74 | -11.09 |
> | w/o L3 | 10.96 | 52.38 | -5.13 |
> | w/o L4 | 36.14 | 60.56 | 10.00 |
> | NO-INFO | 36.00 | 61.34 | -1.00 |
>
> (5) The hyperparameters are different from those for the Norman dataset, but the tuning pipeline remains the same.
>
> > **Q2/W1.2.In the presentation of the method in 3.1, the author(s) fix the proportion between, and (Eq. 4). Could the authors provide the theoretical or empirical justification for this specific modeling choice? A brief ablation study showing that the model is robust to this choice, or that this choice is optimal, would strengthen the clarity of the model design choices here.**
>
> (1) This modeling choice is based on previous research[1].
>
> (2) The Problem (Ambiguity): The original loss function ($\mathcal{L}_{1}$) does not depend on the individual parameters that control uncertainty ($\kappa$ and $\nu$). Instead, it depends on a *combination* of them. This means that infinitely many different combinations of these parameters can produce the exact same loss value, making it impossible for the model to learn the correct, unique uncertainties. This issue is called "degeneration".
>
> (3) The Solution (Fixing the Ratio): To solve this, the authors propose to link, or "couple," the two parameters with a constant ratio, $r$, such that $\nu_{i}=2\kappa_{i}$. This removes the ambiguity and eliminates the need for a separate, flawed regularization term that was used in prior work.
>
> > **Q3.How robust is the model to misspecification of its Normal-Inverse-Wishart distribution? It may be nice to know how strong the regularization of the term is in handling any model misspecification here, since it encourages the pseudo E-distance derived from the Normal-Inverse-Wishart prior to match the training data E-distance. A discussion of its robustness or a simple experiment on synthetic data with a known non-Gaussian ground truth (but still reasonable for single cell settings) to demonstrate that not only the predictions but also the uncertainty scores are robust would clarify the limitations and strengths of this paper.**
>
> This is useful advice, and we conducted the following experiments in response.
>
> In single-cell preprocessing, data is usually first log-normalized. Although raw count data is very sparse, it sometimes contains a few very large counts. Thus, we first normalize to increase differences, then take the logarithm to reduce the long tail. In this way, single-cell data can be treated as having a somewhat Gaussian distribution. However, as you suggested, in a non-Gaussian setting, we did not perform log-normalization. Instead, we used min-max linear normalization, which yields a very asymmetric distribution as our input. Other operations remained the same. We still use the Norman dataset as an example, and the following are the results.
>
> We conducted two trials: the first uses a non-normal input and a non-normal E-distance, while the second uses a non-normal input and a normal E-distance.
>
> | Model | PCC1 | PCC1_DEG | ACC1 | ACC1_DEG |
> | :--- | :--- | :--- | :--- | :--- |
> | Ours(no_normal) | 16.37 | 22.25 | 53.10 | 71.21 |
> | Ours(no_normal_signal_normal) | -47.71 | -15.32 | 27.01 | 36.55 |
>
> | Model | SPCC | ACC2 |
> | :--- | :--- | :--- |
> | Ours(no_normal) | -13.31 | 41.94 |
> | Ours(no_normal_signal_normal) | 10.98 | 41.94 |
>
> The results show that our model cannot handle non-Gaussian distributions. We also note that using an E-distance derived from a normal distribution assumption deteriorates model performance in this setting. We think this may be because the E-distance is a distance calculation suitable for Gaussian distributions. This inspires the idea that the E-distance could be replaced with another distribution distance metric, but that is outside the scope of this paper. However, we sincerely thank you for your advice and will include this limitation regarding the distribution assumption in our paper.
>
> [1] Multivariate Deep Evidential Regression.

---

> > ### Comment · Reviewer_YtU5 · 2025-08-02
> >
> > Thank you. I appreciate your thorough response and transparency with the follow-up experiments. These additions help address my concerns about the choices made for the experiments and model limitations. I think it would be nice to mention that perhaps a similar approach could be adapted to e.g. Poisson distribution for single-cell count data.

---

> > > ### Author Response · Authors · 2025-08-05
> > > **Response Regarding the Model's Distributional Assumption**
> > >
> > > Dear Reviewer YtU5,
> > >
> > > Thank you for your valuable suggestion. We agree that extending our model with a multivariate Poisson distribution is a promising future direction, and we will add this to the discussion in our paper.
> > >
> > > We chose the Gaussian assumption primarily to ensure a fair comparison with existing baselines. For these baselines, performance metrics like PCC are evaluated against continuous, log-normalized ground truth. In contrast, a model using a Poisson distribution would be evaluated against discrete count data. Comparing metrics across these different data types would create an incompatible and unfair evaluation standard.
> > >
> > > We attempted to resolve this by transforming the continuous predictions from the baselines back into the count domain. However, this approach led to a significant degradation in PCC. We attribute this to the non-linear nature of log-normalization, which distorts the correlation and makes a fair comparison challenging.
> > >
> > > Therefore, the Gaussian assumption was the first choice for the comparative analysis conducted in this study.

---

> > > ### Author Response · Authors · 2025-08-07
> > > **Looking forward to your feedback**
> > >
> > > Dear Reviewer YtU5,
> > >
> > > Thank you for your insightful and constructive review. **We found your advised experiment with the model's distributional assumption particularly inspiring, and we appreciate this opportunity to discuss it with you.** Please let us know if you have any further questions or require additional clarification. Thank you again for your time and effort; your feedback has been instrumental in improving our research.
> > >
> > > Best,
> > >
> > > The Authors

---

> > > > ### Author Response · Authors · 2025-08-08
> > > >
> > > > Dear Reviewer YtU5,
> > > >
> > > > To follow up on our discussion on limitations, we present some new results that better demonstrate the potential of PRESCRIBE. **As the discussion period is drawing to a close, we don't expect a further reply.** This comment is simply intended to conclude our discussion.
> > > >
> > > > 1. First, we found that **an appropriate supervised signal can indeed alleviate distributional misspecification**. We replaced the supervised E-distance calculation with Optimal Transport (OT). While this does not separately measure pairwise/self-wise distances, it does not affect the $\mathcal{L}_{3}$ supervisor, which supervises total uncertainty. The pseudo-E-distance calculation (combining entropy and evidence) remains unchanged. This approach yielded promising improvements in both accuracy and uncertainty estimation:
> > > >
> > > > | Model | PCC1 | PCC1_DEG | ACC1 | ACC1_DEG |
> > > > | :--- | :--- | :--- | :--- | :--- |
> > > > | Ours (no_normal) | 16.37 | 22.25 | 53.10 | 71.21 |
> > > > | Ours (no_normal_signal_normal) | -47.71 | -15.32 | 27.01 | 36.55 |
> > > > | Ours (no_normal_signal_ot) | **27.12** | **32.99** | **54.49** | **71.72** |
> > > >
> > > > <br>
> > > >
> > > > | Model | SPCC | ACC2 |
> > > > | :--- | :--- | :--- |
> > > > | Ours (no_normal) | -13.31 | 41.94 |
> > > > | Ours (no_normal_signal_normal) | 10.98 | 41.94 |
> > > > | Ours (no_normal_signal_ot) | **14.31** | **45.16** |
> > > >
> > > > Both the accuracy and the quality of the estimated uncertainty have improved.
> > > >
> > > > 2. Second, after consideration, we think **making our framework compatible with the Poisson distribution seems possible**, as it also belongs to the natural exponential family. This requires some modifications:
> > > > * The Normal-Inverse-Wishart prior is replaced by the Poisson's conjugate prior, the Gamma distribution ($\text{Gamma}(\alpha, \beta)$).
> > > > * The pseudo-E-distance still combines entropy and evidence, but the 'evidence' term $n$ now signifies the parameter $\beta$. Intuitively, a smaller $\beta$ indicates the distribution is closer to the prior.
> > > > * We recommend using OT for the supervised signal in this case.
> > > >
> > > > Thank you again for your time and valuable feedback.
> > > >
> > > > Best,
> > > >
> > > > The authors

---

### Author Response · Authors · 2025-08-08

**General Response**

Dear Reviewers,

We agree and are committed to improving the manuscript's clarity. In the final version, **we will add those high-level descriptions before presenting mathematical or technical details.** This will help ensure our work is accessible to a broader audience.

>**Core Idea: Pseudo E-distance as a unified surrogate for uncertainties arising from both out of distribution (OOD) prediction and randomness stemming from the training data.**

Essentially, the E-distance summarizes both the model's posterior predictive distribution, which is under heavy influence of the randomness from the training data; and the similarity of the prediction to a user-defined OOD “null” distribution, which quantifies how much is the prediction supported by the training data, if it is supported at all.

>**The Prior: We will define the prior as the "default" or "null" state that the model's prediction collapses to in cases of high epistemic uncertainty.**

In our current writing, the default “null” distribution is assigned as the unperturbed control cell state. We acknowledge that this is a minor oversight as it cannot distinguish OOD genes and genes that are supported by strong evidence but has little effect to the cell expression profile. However, this is a minor issue and can be easily corrected by re-assigning the “null” distribution to a distant expression profile that is significantly different from any of the training data, including the unperturbed control state. Furthermore, the “null” distribution is user-changeable and can be set manually based on the study context.

>**Encoder: We will describe the encoder as a module that processes the cell's basal state (c) and the specific perturbation (x) to generate a concise latent embedding.**

We want to emphasize that the encoder is external to our framework and is responsible of reflecting the functional similarity in between genes. We want to emphasize that our framework is agnostic to the choice of the encoder and is compatible with embeddings generated from multiple sources, including ESM, scFoundation, GNN, scGPT, set transformer, etc.

>**Normalizing Flow: This module's role is to estimate the density of the training data in the latent space. It takes the latent embedding as input and outputs the evidence (ν), which quantifies whether the input lies in a familiar (high-density) or unfamiliar (low-density) region of the space.**

Here high-density means that there are many biologically similar genes in the training data that supports the prediction of the unseen gene and low-density vice versa. An OOD gene would be located in a low-density area, which will force the prediction to revert to the pre-defined “null” distribution in the inverse Wishart space.

>**Decoder: The decoder uses the latent embedding to output the initial parameters for the predictive distribution (e.g., mean and variance).**

We will clarify that these are not the final posterior predictions but rather inputs for the final update step.

>**Bayesian Update: This is the final step where the evidence (ν) from the normalizing flow, the predictive parameters from the decoder, and the distance to the OOD “null” distribution are combined.**

This step produces the final posterior distribution and our pseudo-E-distance uncertainty score.

Finally, we want to emphasize that in scientific settings beyond multi-omics data analysis, such as in material science, it is quite common that a machine learning model is being constantly queried with unseen predictions. It is also quite common that the training data themselves are experimental observations that are riddled with uncertainties. We believe our framework, albeit currently focused on gene perturbation prediction, has a greater applicability to a broad scenarios in AI for science.

---

### Author Response · Authors · 2025-08-09
**Final Thoughts**

We thank all reviewers for their thoughtful and constructive feedback. In response, we have carefully considered each point raised and have made significant revisions to the manuscript.

**Key revisions**, guided by the reviewers' suggestions, include the following:

* We will clarify the distribution-specific feature of our rank loss design and discuss how our model can be extended to support other exponential family distributions (e.g., Poisson), as suggested by Reviewer YtU5.
* We will clarify the objective of this work, as suggested by Reviewer HczP and UDmR.
* We will add high-level descriptions of our method and adopt new metric notations to attract a broader audience, as advised by UDmR.
* We will incorporate the ECE calibration metric, as suggested by Reviewer tFuR.
* We will add ablation studies on model embeddings, update the format of all results, include statistical significance tests, and expand our related work section, as suggested by Reviewer Z96M.

---

Perturbation prediction often involves predicting effects on genes unseen during training. Consequently, the efficacy of the prediction jointly depends on 1) how similar the target gene is to any of the perturbed genes covered by the training data (model/epistemic uncertainty) and 2) the quality of such similar-to-target training data (data/aleatoric uncertainty), if there exists any. Both are important in determining the prediction reliability, as gene perturbation is a highly stochastic biochemical process.

To briefly summarize our work, we propose an uncertainty-aware framework for single-cell perturbation prediction. We address a critical challenge in predicting gene perturbation effects on cells: estimating the reliability and the efficacy of the prediction. **Our core contribution is to use pseudo-E-distance (distance from prior and entropy) as a surrogate to jointly measure both epistemic (model) uncertainty and aleatoric (data) uncertainty.** Our results show that our model leads to improved uncertainty calibration and demonstrates practical utility. We envision future extensions could support more exponential family distributions, and our model's performance could be further enhanced by leveraging state-of-the-art perturbation embeddings.

---

We believe these changes will strengthen the manuscript and clarify the contributions of our work. We appreciate the opportunity to improve our paper and are confident that the revised version better reflects the significance and originality of our research.

Thank you again for your time and effort in reviewing our submission. We look forward to your final decision.

---

### Decision · Program_Chairs · 2025-09-17

**Decision:**

Accept (poster)

**Comment:**

**(a) Summary**
This paper introduces PRESCRIBE, a framework for predicting single-cell transcriptional responses to genetic perturbations while providing calibrated uncertainty estimates. The method extends the Natural Posterior Network (NatPN) to the multivariate case with a Normal-Inverse-Wishart prior, introduces a pseudo E-distance metric to jointly capture epistemic and aleatoric uncertainty, and evaluates the approach on three benchmark single-cell perturbation datasets. The authors provide ablation studies, calibration metrics, and transfer experiments, positioning PRESCRIBE as a practically useful tool for uncertainty-aware perturbation modeling.

**(b) Strengths**
- Addresses a critical gap in single-cell perturbation prediction by quantifying prediction uncertainty, an important step for practical applications in biology and drug discovery.
- Provides a theoretically motivated extension of NatPN to the multivariate setting and carefully derives a pseudo E-distance surrogate for uncertainty.
- Backed by extensive experiments across multiple datasets, including calibration analyses, ablations on model components and embeddings, and a transfer task across cell types.
- Author responses were thorough, with additional experiments (e.g., use of alternative embeddings, calibration with ECE, statistical significance tests) that strengthened the empirical case.
- Reviewers generally acknowledged the technical soundness and practical significance of the contribution.

**(c) Weaknesses**
- The paper remains technically dense and will require improvements in exposition for accessibility to both ML and biology audiences.
- Some methodological choices, such as the Gaussian assumption and encoder design, could have been further motivated relative to alternatives like Poisson-based models or set transformers.
- Calibration improvements, while demonstrated, are incremental in some metrics compared to strong baselines (e.g., GEARS with modified uncertainty scores).
- Generalization beyond the three main datasets is not fully explored, leaving open questions about broader applicability.

**(d) Key reasons for decision**
Most reviewers were ultimately supportive, with three (YtU5, HczP, tFuR) recommending acceptance and one (UDmR) moving from borderline reject to a borderline accept after the rebuttal. Reviewer Z96M maintained reservations, but the author rebuttal provided additional ablations, specifications, and statistical tests that addressed many concerns. The consistent trajectory of the reviews and the quality of the rebuttal indicate that the work represents a meaningful contribution despite clarity issues. The core methodological advance, uncertainty-aware prediction of perturbation responses, is judged to be significant enough to merit acceptance.

**(e) Discussion and changes**
The author–reviewer discussion was constructive and led to concrete improvements. Reviewers’ concerns about clarity, scope of ablations, and calibration metrics were met with new experiments (e.g., encoder ablations, cross-cell-type transfer, ECE metrics, statistical tests). While Reviewer Z96M remained less convinced, others expressed satisfaction with the responses and confirmed that their concerns were resolved. The rebuttal strengthened the empirical case and clarified limitations, particularly regarding distributional assumptions and generalization. Overall, the balance of the discussion shifted positively, supporting an acceptance decision.

**Decision: Accept (Poster).**
Overall justification: The paper presents a well-motivated and empirically supported contribution addressing uncertainty quantification in single-cell perturbation prediction, with sufficient strengths to outweigh remaining concerns about clarity and scope.